# Natural variations of adolescent neurogenesis and anxiety predict the hierarchical status of adult inbred mice

Fabio Grieco [iD][1], Atik Balla [iD][1], Thomas Larrieu [iD][1,2] ✉ & Nicolas Toni [iD][1,2] ✉

## Abstract

**Hierarchy provides a survival advantage to social animals in challenging circumstances. In mice, social dominance is associated with trait anxiety which is regulated by adult hippocampal neurogenesis. Here, we test whether adolescent hippocampal neurogenesis may regulate social dominance behavior in adulthood. We observe that adolescent individuals with higher trait anxiety and lower levels of hippocampal neurogenesis prior to the formation of a new group become dominants, suggesting that baseline adolescent neurogenesis predicts hierarchical status. This phenotype persists beyond social hierarchy stabilization. Experimentally reducing neurogenesis prior to the stabilization of social hierarchy in group-housed adolescent males increases the probability of mice to become dominant and increases anxiety. Finally, when innate dominance is assessed in socially isolated and anxiety-matched animals, mice with impaired neurogenesis display a dominant status toward strangers. Together, these results indicate that adolescent neurogenesis predicts and regulates hierarchical and situational dominance behavior along with anxiety-related behavior. These results provide a framework to study the mechanisms underlying social hierarchy and the dysregulation of dominance behavior in psychiatric diseases related to anxiety.**

**Keywords** Social Hierarchy; Adult Neurogenesis; Hippocampus; Anxiety; Situational Dominance
**Subject Category** Neuroscience

## Introduction

Hierarchical structures in animal societies arise to address environmental challenges essential for group survival: Hierarchies help manage resource fluctuations, reduce conflicts, conserve energy, and foster stability, providing a significant survival advantage. Although high social status can buffer stress in stable hierarchies, high-status individuals experience heightened stress in times of instability (Knight and Mehta, 2017;

Sapolsky, 2005). In humans, a number of psychopathologies are linked to disruptions in social hierarchy: externalizing disorders like mania-proneness and narcissistic traits are associated with increased dominance behavior, while anxiety and depression associate with subordination and submissiveness (Neumann et al, 2010; Tang-Smith et al, 2015). In mice, dominant individuals display increased anxiety-related behavior and depressive-like symptoms as compared to subordinates (Cherix et al, 2020; Larrieu et al, 2017; Larrieu and Sandi, 2018) and it is unclear whether dominance status results from or induces trait anxiety. While social dominance in male mice is established before weaning and remains stable throughout adulthood (Lindzey et al, 1961; Wang et al, 2011; Zhou et al, 2018), there is a sensitive period for experience-dependent social dominance plasticity during adolescence that depends on mechanisms of cortical plasticity (Bicks et al, 2021). Thus, mechanisms of brain plasticity occurring in the adolescent brain may regulate social dominance behaviors that were learned during the first stages of life.

The most drastic form of postnatal brain plasticity, adult neurogenesis, consists in the continuous formation of neurons and their integration into pre-existing circuits (Altman and Das, 1965). In the hippocampus, adult neurogenesis plays a role in mechanisms of memory and mood regulation (Anacker and Hen, 2017) and inhibiting adult hippocampal neurogenesis (AHN) increases anxiety-like behavior (Revest et al, 2009) and stress vulnerability (Anacker et al, 2018; Hill et al, 2015; Snyder et al, 2011; Surget et al, 2011). Given that dominant individuals show emotional alterations, we hypothesized the existence of a link between AHN, trait anxiety, and dominance behavior and tested this link using observational and interventional studies at the interface between late adolescence and adulthood.

## Results and discussion

### Segregation of naive C57BL/6J inbred mice into dominant and subordinate populations

We investigated the relationship between social hierarchy, adolescent hippocampal neurogenesis, and anxiety. Six-week-old adolescent male mice were individually housed and tested for trait anxiety using an elevated plus maze (EPM), light-dark test (LDT), and open field test (OFT). One day later, to assess baseline adolescent neurogenesis, mice received 5-Iodo-2′-Deoxyuridine (IdU) injections to label proliferating

[1]Center for Psychiatric Neuroscience, Department of Psychiatry, Lausanne University Hospital, University of Lausanne, Prilly, Switzerland. [2]These authors contributed equally: Thomas Larrieu, Nicolas Toni. ✉E-mail: thomas.larrieu@chuv.ch; nicolas.toni@unil.ch

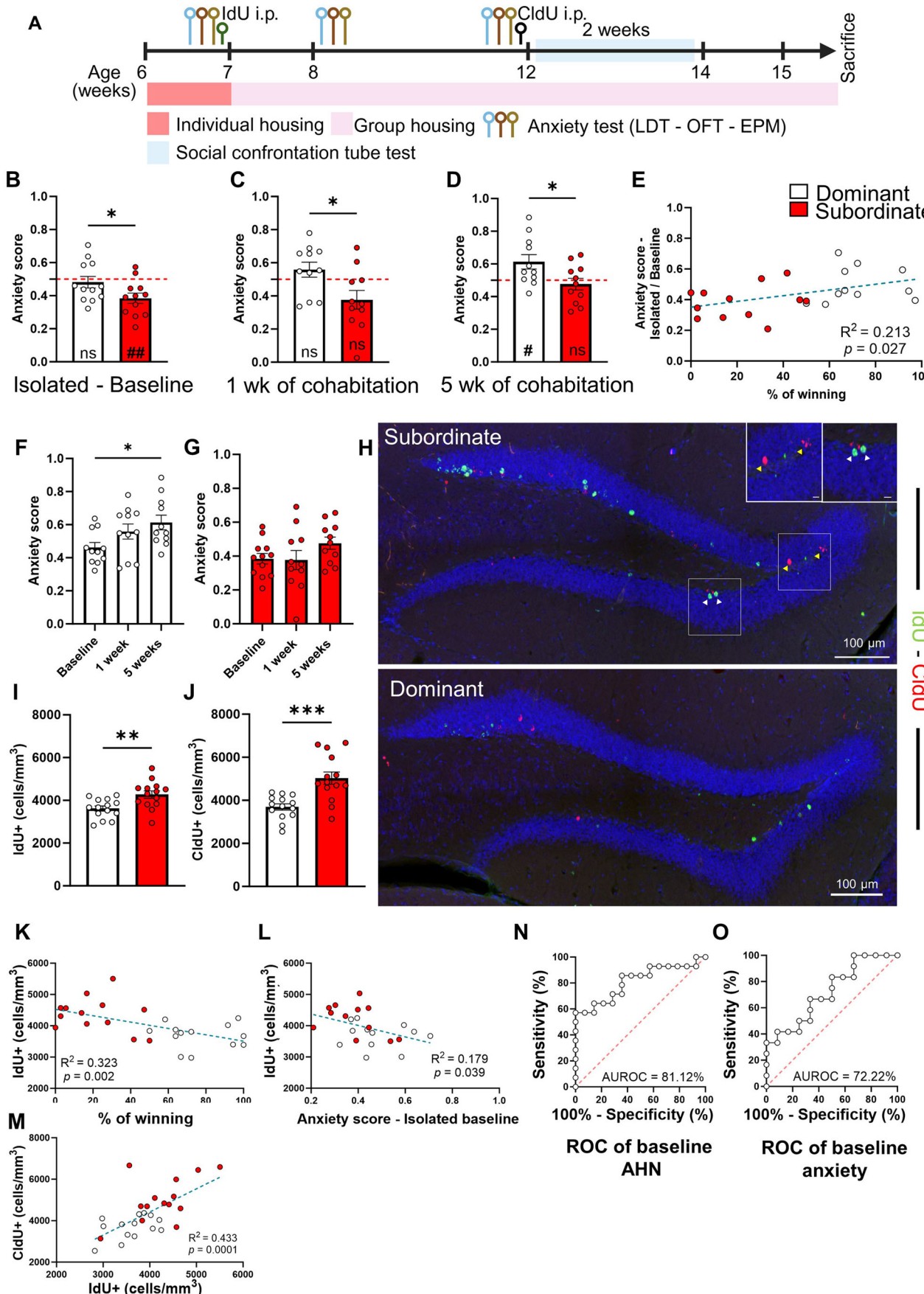

◀

**Figure 1. Dominant mice exhibit persistent anxiety traits and reduced adolescent and adult hippocampal neurogenesis (AHN).**

(A) Experimental design illustrating the evaluation of anxiety, AHN, and social rank. (B–D) Anxiety scores in dominant and subordinate mice during isolation (baseline, B) and after 1 (C) and 5 weeks (D) of cohabitation. Anxiety was normalized based on time spent in the dark chamber (LDT), closed arm (EPM), and thigmotaxis (OFT). (B) Baseline anxiety ($t22 = 2.07$, $p = 0.5$, unpaired t-test, $n = 12$ mice per group). Dominants: $t11 = 0.53$, $p = 0.602$; Subordinates: $t11 = 3.71$, $p = 0.003$, one-sample t-test as compared to the population mean (dashed line). (C) Anxiety after 1 week of cohabitation ($t20 = 2.52$, $p = 0.02$, unpaired t-test, $n = 11$ mice per group). Dominants: $t10 = 1.29$, $p = 0.225$; Subordinates: $t10 = 2.20$, $p = 0.052$, one-sample t-test. (D) Anxiety after 5 weeks of cohabitation ($t20 = 2.38$, $p = 0.027$, unpaired t-test, $n = 11$ mice per group). Dominants: $t10 = 2.54$, $p = 0.02$; Subordinates: $t10 = 0.65$, $p = 0.525$, one-sample t-test. (E) Correlation between baseline anxiety score and percentage of winning during SCTT ($R^2 = 0.2133$, $p = 0.027$, simple linear regression). (F, G) Anxiety evolution in dominants (F) and subordinates (G) (F, F2,30 = 3.580, $p$ 0.04, one-way ANOVA, $n = 11$ mice per group. G, F2,31 = 1.717, $p = 0.196$, one-way ANOVA, $n = 11$ mice per group). (H) Confocal projections of the dentate gyrus showing IdU (green, white arrowheads), CldU (red, yellow arrowheads), and DAPI (blue) staining in subordinates and dominants. (I, J) Quantification of IdU-positive cells (I, $t26 = 3.101$, $p = 0.005$, unpaired t-test, $n = 14$ mice per group) and CldU-positive cells (J, $t26 = 4.15$, $p = 0.0005$, Welch's test, $n = 14$ mice per group). (K, L) Correlation of IdU counts with winning percentage (K, $R^2 = 0.323$, $p = 0.002$) and baseline anxiety score (L, $R^2 = 0.179$, $p = 0.039$; simple linear regression). (M) Correlation between IdU+ cells and CldU+ cells (R2 = 0.433, $p = 0.0001$, simple linear regression. (N, O) ROC curve analysis for discriminating dominant mice based on baseline adolescent neurogenesis (N) and anxiety (O). Histograms show average ± SEM; *$p < 0.05$, **$p < 0.01$, ***$p < 0.001$. ns = not significant. The dashed line in the anxiety score represents the population mean and is used to assess within-group anxiety using a one-sample t-test where #: $p < 0.05$, ##: $p < 0.01$. Scale bar insets: 10 μm. Source data are available online for this figure.

cells (Fig. 1A). They were then randomly assigned to cages of four and retested for anxiety after one week of cohabitation, corresponding to the active formation of social hierarchies. The mice were left undisturbed for an additional four weeks (i.e., for a total of 5 weeks of cohabitation), which is sufficient to enable the establishment of a stable hierarchy (Larrieu et al, 2017), and anxiety was assessed again at 11 weeks of age (adulthood). To examine adolescent hippocampal neurogenesis, 5-Chloro-2′-deoxyuridine (CldU) was administered 24 h after the last anxiety testing. Social rank was determined using a social confrontation tube test (SCTT) over two weeks in a round-robin design. The SCTT is a simple behavioral test that assesses dominance in a cage, reliably replacing other measures such as the warm spot test, territory urine marking or ultrasound vocalizations (Fan et al, 2019). In an independent cohort, hierarchical stabilization was already observed after 4 days (Fig. EV1A–D). The rank in SCTT was unrelated to time spent in the tube during habituation (Fig. EV1E) or body weight before and after testing (Fig. EV1F–H) as reported in prior studies (Cherix et al, 2020; Larrieu et al, 2017). For analysis, ranks 1–2 were categorized as dominant, and ranks 3–4 as subordinate.

### Future dominant mice display heightened trait anxiety compared to subordinates

Observations of animals exposed to an anxiogenic context (Grasmuck and Desor, 2002) or following treatment with anxiolytics (Schroeder et al, 1998) suggest that anxiety may increase dominance behavior. We observed that before cohabitation, mice that later became dominant exhibited higher anxiety score than subordinate mice (Fig. 1B). This was reflected by increased time spent in the closed arms and reduced time spent in the open arms of an EPM, prolonged exploration of the dark compartment in a LDT, and increased thigmotaxis in an OFT (Fig. EV2A–G). Notably, comparable findings were obtained after 1 and 5 weeks of cohabitation (Figs. 1C,D and EV2H–S), suggesting that the anxiety profile observed in high-ranking mice is not solely attributable to the active agonistic behavior during the first week of hierarchy establishment but instead results from a pre-existing, individual anxiety trait. However, the correlation between anxiety and the proportion of wins in the SCCTT decreased over time and disappeared after hierarchy stabilization (after 5 weeks of cohabitation; Figs. 1E and EV3A,B), suggesting that social dominance may be independent from anxiety.

When comparing the three time points (baseline and after 1 and 5 weeks of cohabitation), anxiety levels in dominants increased over time, as revealed by a one-sample t-test analysis comparison with the 0.5 value of anxiety score, while the low anxiety observed in subordinates remained stable (Fig. 1F,G). This increased anxious phenotype appearing during the cohabitation period supports the view that the ongoing maintenance of a social status, even within stable social hierarchies, imposes continuous stress that could be perceived as more salient in dominant individuals than subordinates (Sapolsky, 2005). Together, these observations suggest a bidirectional regulation between anxiety and dominance, where mice with higher anxiety score before group formation dominate mice with lower anxiety score and that after group formation, dominant status may induce anxiety.

### Future dominant mice display lower levels of baseline adolescent hippocampal neurogenesis

The relationship between adult neurogenesis and social dominance varies across species, sexes, and methodologies (Cope et al, 2020; Holmes, 2016; Horii et al, 2017; Karamihalev et al, 2020; Kozorovitskiy and Gould, 2004; Opendak et al, 2016; Palle et al, 2019; Wu et al, 2014). In rats, studies using the visible burrow system found higher neurogenesis in dominant individuals (Kozorovitskiy and Gould, 2004; Opendak et al, 2016), while dominant naked mole-rat breeders showed reduced DCX expression in the dentate gyrus (Peragine et al, 2014). In mice, studies using home-cage agonistic behaviors (Horii et al, 2017) or the social confrontation tube test (Palle et al, 2019) reported no clear differences in neurogenesis between dominant and subordinate individuals. To determine if hierarchy correlates with hippocampal neurogenesis, we analyzed subgranular zone cells labeled with IdU (cells that proliferated during adolescence) or CldU (cells that proliferated during adulthood). Dominant individuals exhibited significantly fewer IdU- and CldU-positive cells than subordinates (Fig. 1H–J). The number of IdU-labeled cells inversely correlated with the proportion of wins (Fig. 1K) and with anxiety at baseline (Fig. 1L) but not after one or five weeks of cohabitation (Fig. EV3C,D). Similarly, the number of CldU-labeled cells inversely correlated with the proportion of confrontations won (Fig. EV3E) and positively correlated with the number of IdU-labeled cells (Fig. 1M). Finally, a ROC curve analysis showed that adolescent hippocampal neurogenesis distinguished dominant from subordinate adults with 81% accuracy (Fig. 1N) and baseline anxiety achieving 72% accuracy (Fig. 1O). Together, these findings indicate that

natural variations in adolescent hippocampal neurogenesis predict hierarchical rank in a newly formed group, with lower baseline adolescent neurogenesis predisposing individuals to dominance behavior in adulthood.

## Inhibiting neurogenesis in late adolescence increases social dominance behavior and anxiety

Social dominance is expressed by either hierarchical (confrontation between cage mates) or situational (confrontation between strangers) dominance behavior (Taki et al, 2020; Wang et al, 2014). To assess the role of adolescent neurogenesis in hierarchical dominance behavior, mice were housed in groups of 4 individuals. Starting after 2 days of cohabitation, while the social hierarchy was still being established, two of the four mice were randomly treated either with temozolomide (TMZ), an antimitotic drug or vehicle (saline, NaCl 0.9%) for the first 3 days of each week for four consecutive weeks, followed by 2 weeks of rest. Although TMZ was injected systemically and may therefore affect peripheral mechanisms, this treatment has been shown to impair adult neurogenesis (Egeland et al, 2017; Garthe et al, 2009). To assess cell proliferation, mice were injected with BrdU after the first week of TMZ administration. After this, mice were tested in the SCTT using a round robin design, followed by anxiety assessments (Fig. 2A). Consistent with previous studies (Egeland et al, 2017; Garthe et al, 2009), the TMZ treatment reduced the number of BrdU-positive cells (Fig. 2B,C). Furthermore, mice subjected to TMZ treatment demonstrated a significant increase in the probability of acquiring a dominant status in the SCTT (Fig. 2D) and a significant increase in the number of wins as compared to NaCl-treated mice (Fig. 2E,F). Finally, we found that TMZ-treated mice showed increased anxiety score as compared to NaCl-treated mice (Figs. 2G and EV4). Thus, inhibiting adolescent neurogenesis increases anxiety and social hierarchy in the home cage in adult individuals.

## Inhibiting neurogenesis in late adolescence increases situational social dominance behavior in anxiety-matched conditions

TMZ affects neurogenesis in the olfactory bulb and previous studies showed that the detection of odors or pheromones are crucial for social recognition and memory (Cope et al, 2020; Garcia-Gomez et al, 2022; Garrett et al, 2015; Pereira-Caixeta et al, 2018; Palle et al, 2020; Palle et al, 2019). Thus, a reduction of neurogenesis in the subventricular zone may affect social memory and stimulate the appearance of novel agonistic behavior which could participate to the effect of TMZ on dominance behavior. To assess the role of adolescent neurogenesis in absence of social memory, we tested the effect of TMZ in a second cohort of male mice which were singly housed. Furthermore, situational dominance seems ethologically more relevant than hierarchical dominance observed in individuals sharing a cage, since wild mice live in territories occupied by a single male, several females and their progeny, and therefore, male mice establish dominance upon their first encounter (Kappel et al, 2017).

One week after arrival, mice were weighed and assessed for trait anxiety using an elevated plus maze, a light-dark test and an open field test. The combined anxiety score was used to assign mice to either the TMZ or the vehicle group to match weight and anxiety between groups. Mice were then treated with either TMZ or vehicle for 4 weeks, followed by 4 weeks of resting period. After this, their trait anxiety was assessed

using an open-field, an elevated plus maze and a light-dark tests. TMZ-treated mice were then paired with vehicle-treated mice of similar anxiety traits and body weight and confronted in the SCTT for 5 days, twice a day (morning and afternoon, Fig. 3A). We found that across all sessions, TMZ-treated animals exhibited an increased probability of becoming dominant (Fig. 3B) and an increased number of total wins and wins per day as compared to vehicle-treated animals (Fig. 3C,D). Although the anxiety score was similar between groups (Figs. 3E and EV5), the anxiety score of the vehicle group was higher than in group-housed mice (Fig. 2G), likely reflecting the anxiogenic effect of long-term social isolation (Grigoryan et al, 2022; Ieraci et al, 2016). Thus, in absence of social memory and differences in trait anxiety, inhibiting neurogenesis in adolescence and early adulthood increases social hierarchy and dominance behavior in a situational context.

In this study, we examined the role of adolescent hippocampal neurogenesis in hierarchy formation. While we found that inhibiting neurogenesis in late adolescence/early adulthood increased the hierarchical status of mice that had never met before, it remains to be determined whether the effect of neurogenesis inhibition extends beyond the critical period of plasticity in dominance behavior (Bicks et al, 2021). Several brain regions are known to be involved in the perception and learning of social dominance, such as the nucleus accumbens (Hollis et al, 2015; Larrieu et al, 2017), the ventral tegmental area (van der Kooij et al, 2018) and the prefrontal cortex (Wang et al, 2014; Wang et al, 2011). In particular, the medial prefrontal cortex (mPFC) seems to play a predominant role in social hierarchy behavior: the winning history of individual mice remodels thalamic connections to the mPFC, leading to long-term modifications of the dominance status (Zhou et al, 2017). Furthermore, the excitatory synaptic efficacy in the mPFC is higher in dominant mice than in subordinates, and the bidirectional manipulation of synaptic efficacy in the mPFC alters social dominance (Wang et al, 2011; Zhou et al, 2017). Adult neurogenesis is a particularly drastic mechanism of plasticity that enables a rapid adaptation to novel living conditions. Interestingly, the ventral hippocampus projects to the mPFC and the activity of these projections is involved in social memory (Phillips et al, 2019) as well as in the response to anxiogenic experience (Adhikari et al, 2010). Owing to their specific connectivity, immature hippocampal neurons decrease the activity of mature neurons from the ventral dentate gyrus in a stressful environment (Anacker et al, 2018). Thus, one may speculate that reduced hippocampal neurogenesis may promote dominance behavior by increasing the output from the ventral hippocampus to the mPFC.

Due to their crucial role in individual survival, allocation of territory or access to reproduction and food resources, dominance behaviors strongly influence the social organization and structure of a group. The reduced neurogenesis in dominant mice may confer behavioral advantage in their natural habitat: In the wild, *Mus musculus* live in demes formed by one dominant adult male that defends the deme's territory, several females, and their offspring. As soon as the male offsprings reach sexual maturity, they will either remain in the deme under the dominance of the adult male until they inherit the territory or leave the original deme to form their own (Brust et al, 2015; Latham and Mason 2004). In the framework of these naturalistic considerations, one can speculate that individuals with higher neurogenesis will display lower anxiety and are therefore more likely to explore new territories to form their own deme or to identify new food source. In contrast, individuals with lower neurogenesis and higher trait anxiety may remain in and defend the deme's territory, a strategy for which dominance behavior provides an advantage. In this

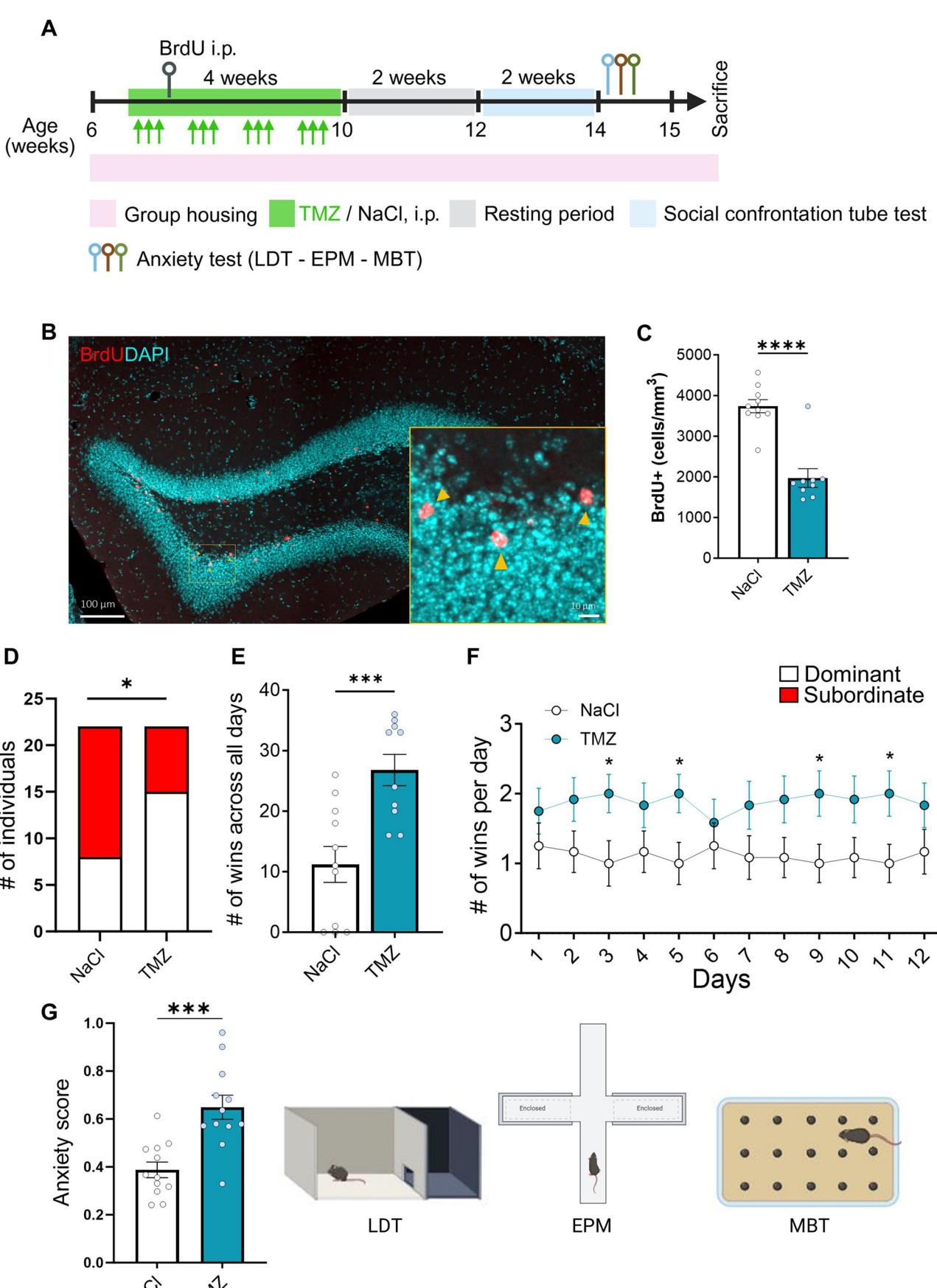

**Figure 2. Adolescent neurogenesis depletion increases hierarchical dominance.**

(A) Experimental design for TMZ administration, anxiety assessment and SCTT of group-housed mice. (B) Representative confocal microscopy image of the dentate gyrus immunostained for BrdU (red cells highlighted with yellow arrows) and DAPI (blue). (C) BrdU-positive cell density in the dentate gyrus ($t_{17} = 6.42$, $p = 0.000006$, unpaired t test, two-tailed $n = 10$ mice per TMZ group and 9 per NaCl). (D) Contingency table for the effect of TMZ treatment on dominance behavior (Chi-square test, two-sided, $p = 0.0346$, $n = 21$ subordinate, $n = 23$ dominants). (E) Histogram representing the number of wins in SCTT across all days ($t_{19} = 3.92$, $p = 0.0009$, unpaired t test, two-tailed, $n = 11$ mice for NaCl and $n = 10$ mice for TMZ). (F) SCTT outcomes day by day in number of wins (two-way ANOVA: interaction effect: $F_{11,242} = 1.53$, $p = 0.104$, time effect: $F_{11,242} = 0.12$, $p = 0.99$, mouse effect: $F_{1,22} = 3.58$, $p = 0.071$, subject effect: $F_{22,242} = 69.84$, $p = 0.00000001$; $n = 12$ mice per group; in Day 3: $t_{22} = 2.34$, $p = 0.028$, unpaired t test, two-tailed, $n = 12$ mice per group; in Day 5: $t_{22} = 2.44$, $p = 0.023$, unpaired t test, two-tailed, $n = 12$ mice per group; in Day 9: $t_{22} = 2.34$, $p = 0.028$, unpaired t test, two-tailed, $n = 12$ mice per group; in Day 11: $t_{22} = 2.345$, $p = 0.028$, unpaired t test, two-tailed, $n = 12$ mice per group). (G) Histogram of the anxiety score derived from LDT, EPM and MB tests, of TMZ and NaCl-treated mice and illustrations of the tests used to assess anxiety ($t_{22} = 4.36$, $p = 0.0002$, unpaired t test, two-tailed, $n = 12$ mice per group. Histograms show average ± SEM, *$p < 0.05$, ***$p < 0.0002$, ****$p < 0.0001$. Source data are available online for this figure.

context, natural variations in hippocampal neurogenesis in late adolescence may be instrumental in diversifying the strategy of the male offspring and therefore, increase their survival. This possibility raises the question of the origin of interindividual differences in hippocampal neurogenesis in inbred strains of mice. While their implication in dominance behavior is still unclear, early life experience such as maternal care, competition between pups or exploration constitute a non-shared environment that may underlie the formation of individual traits leading to dominance or subordination later in life (Freund et al, 2013; Grieco et al, 2024; Kempermann, 2019).

# Methods

### Reagents and tools table

| Reagent/Resource | Reference or Source | Identifier or Catalog Number |
|---|---|---|
| **Experimental models** | | |
| C57BL/6J (*M. musculus*) | Janvier Laboratories (France) | |
| **Recombinant DNA** | | |
| **Antibodies** | | |
| Rat anti-BrdU/CldU | Abcam | ab6326 |
| Mouse anti-IdU | Abcam | ab181664 |
| Alexa Fluor 488 Goat anti-mouse | Invitrogen | A11029 |
| Alexa Fluor 594 Goat anti-rat | Invitrogen | A11007 |
| **Oligonucleotides and other sequence-based reagents** | | |
| **Chemicals, Enzymes and other reagents** | | |
| 5-Iodo-2-deoxyuridine | Sigma-Aldrich | 54-42-2 |
| 5-Chloro-2-deoxyuridine | Sigma-Aldrich | 50-90-8 |
| 5-Bromo-2-deoxyuridine | Sigma-Aldrich | 59-14-3 |
| Temozolomide | Sigma-Aldrich | 85622-93-1 |
| **Software** | | |
| ANY-maze v 7.0 | https://www.any-maze.com/ | |
| GraphPad v 10.0.0 | https://www.graphpad.com/ | |
| **Other** | | |

## Experimental designs

### Experiment 1

We investigated the possible interplay between adolescent hippocampal neurogenesis, anxiety and dominance behavior. Six-week-old adolescent male mice were isolated after their arrival, after three days of acclimatization we tested their basal anxiety level using three different tests, with an interval of 24 h between each test: elevated plus maze (EPM), light-dark test (LDT) and open field test (OFT). After the last test, at postnatal day 48, the mice were injected with 5-Iodo-2′-Deoxyuridine (IdU) to evaluate baseline levels of adolescent hippocampal neurogenesis. After one week of single housing, we formed new anxiety matched groups of 4 animals per cage. We repeated the anxiety tests (EPM, LDT and OFT) one week after the formation of the new group to observe if the establishment of the social hierarchy was influencing anxiety. After a total of 5 weeks of cohabitation, when the social hierarchy is well-established and can be considered stable, we evaluated anxiety levels using the same battery of tests (EPM, LDT and OFT). At the end of the last test, at postnatal day 83, mice were injected with 5-Chloro-2′-deoxyuridine (CldU). After five weeks of cohabitation with their cage mates, mice underwent a social confrontation tube test (SCTT) to determine individual social rank within the cage for 3 weeks. The experimenter was blind to the identity of the mice during data collection and analyses. After the end of all behavioral tests, animals were sacrificed in the morning and their brains dissected out and prepared for histology.

### Experiment 2

We investigated the role of adolescent neurogenesis in the establishment of social hierarchy in a group of four six-week-old male mice. After two days of cohabitation, TMZ was randomly administered to 2 mice from the cage while the other two received NaCl. One week following the initiation of TMZ treatment, mice received an intraperitoneal injection of BrdU to evaluate adolescent neurogenesis in the DG at the end of the experiment. After four weeks of TMZ treatment and an additional 2 weeks of resting period, the mice underwent a SCTT to assess individual social rank. Four weeks after the completion of the TMZ treatment, with an interval of 24 between each test we evaluated anxiety using a LDT, an EPM, and a MBT. The experimenter was blind to the identity of the mice during data collection and analyses. Twenty-four hours after the last behavioral test, the mice were sacrificed, and their brains were removed after perfusion and stored at −20 °C for further processing.

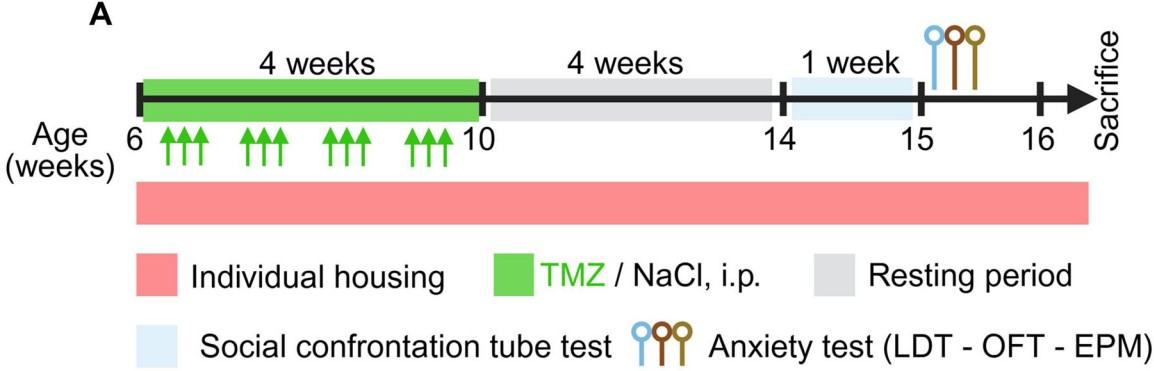

Figure panels B through E

**Experiment 3**

We investigated the role of adolescent neurogenesis in innate social dominance behavior in a situational dominance context. Six-week-old male mice were individually housed upon arrival to prevent the

development of a hierarchical structure and social memory. After the acclimatization week, mice underwent four weeks of TMZ treatment as previously described, followed by a four-week resting period. Subsequently, animals were tested for dominance using a

**Figure 3. Adolescent neurogenesis depletion increases situational dominance.**

(A) Experimental design for TMZ administration, anxiety assessment and SCTT in individually housed animals. (B) Contingency table for correlation between tube test ranks and solution administrated. (Chi-square, two-sided, $p = 0.073$, $n = 10$ mice per group). (C) Histogram representing the number of wins in SCTT across all days ($t18 = 2.12$, $p = 0.0480$, unpaired t test, two-tailed, $n = 10$ mice per group). (D) SCTT outcomes day by day in number of wins. (two-way ANOVA: interaction: $F4,72 = 0.94$, $p = 0.441$; time effect: $F2,111 = 0.00$, $p = 1.00$; treatment effect: $F1,18 = 4.50$, $p = 0.048$; subject effect: $F18,72 = 33.68$, $p = 0.00000001$; $n = 10$ mice per group; in Day 2: $t18 = 2.12$, $p = 0.048$, unpaired t test, two-tailed, $n = 10$ per group; in Day 3: $t18 = 2.63$, $p = 0.016$, unpaired t test, two-tailed, $n = 10$ mice per group). (E) Histograms of the anxiety score derived from OFT, LDT and EPM tests and illustrations of the tests used to assess anxiety ($t18 = 0.335$, $p = 0.741$, unpaired t test, two-tailed, $n = 10$ mice per group). Histograms show average ± SEM, $*p < 0.05$. ns = not significant. Source data are available online for this figure.

SCTT. Twenty-four hours after the final confrontation in the SCTT, mice were subjected to behavioral tests every 24 h, including OFT, LDT, and EPM, to evaluate their anxiety levels. After completion of all behavioral tests, mice were sacrificed, and their brains were removed after perfusion with 4% paraformaldehyde. The brain tissue was then stored at -20 °C for further processing.

## Animals

All experiments were performed according to the ARRIVE guidelines (Percie du Sert et al, 2020) and with the approval of the local Veterinary Authorities (Vaud, Switzerland, authorization number: VD3728) and carried out in accordance with the European Communities Council Directive of 24 November 1986 (86/609EEC). All experiments were performed on C57Bl/6J male mice obtained from Janvier Laboratories (France). Animals were weighted upon arrival as well as weekly to monitor health. Mice were maintained under standard housing conditions on corncob litter in a temperature- ($23 ± 1$ °C) and humidity- (40%) controlled animal room with a 12 h light/dark cycle (8h00–20h00), with unlimited access to food and water. All tests were conducted during the light period.

## TMZ and BrdU/CldU/IdU administration

To assess the basal level of adolescent hippocampal neurogenesis IdU (57.5 mg/kg; 10 mg/ml) was dissolved in a 0.2 N NaOH/saline solution and injected i.p., three times the same day, starting from 11 am with 2 h interval between each injection. Then, mice were injected with a CldU solution (42.5 mg/kg; 10 mg/ml i.p.), three times the same day with 2 h interval between each injection. To reduce neurogenesis, mice were treated with TMZ as previously described (Carrard et al, 2021; Egeland et al, 2017). An intraperitoneal (i.p.) injection of either TMZ (25 mg/kg; 2.5 mg/ml in 0.9% NaCl) or saline (0.9% NaCl) was given for the first three days of the week for a total of 4 weeks, followed by another 4 weeks of resting period between the last injection and the following behavioral test. Moreover, in preparation for the assessment of cell survival via immunostaining, the mice were injected with BrdU after the first cycle of TMZ. Each animal was injected with BrdU (100 mg/kg; 10 ml/kg i.p.) three times the same day, starting from 11 am with 2 h interval between each injection.

## Behavioral tests

### Elevated plus maze test

The test was conducted as previously described (Larrieu et al, 2017). Briefly, the animals were placed in a maze made from black PVC with a white floor. The EPM consists of 4 arms (30 × 5 cm) arranged in a plus shape, with walls on two opposing arms, while the other two arms are exposed to the height of the apparatus (65 cm). The arms converge at the center, forming a small, squared platform (5 × 5 cm). Throughout the tests, the light conditions were maintained stable. The open arms and the center of the maze had a luminous flux of 12–15 lux, while the closed arms had reduced light intensity with 5 lux. The animals were gently introduced into the maze, facing the wall at the end of the closed arms, and were allowed to freely move within the maze for a duration of 5 min. Video recordings of the mice's behavior were captured from above the arena, and tracking analyses were performed using the ANY-maze software to determine the time spent in both the open and closed arms.

### Open-field test

The OFT was conducted as previously described (Larrieu et al, 2017) in a rectangular arena (50 × 50 × 40 cm³) illuminated with dimmed light (30 lux). Mice were introduced near the wall of the arena and allowed to explore for 10 min. Analyses were performed using ANY-maze tracking software by drawing a virtual zone (15 × 15 cm²) in the center of the arena defined as the anxiogenic area. Several parameters were analyzed, including the total distance traveled and the time spent in the different zones.

### Light-dark test

As previously described (Nasca et al, 2015), the apparatus utilized for the LDT consisted of a white wooden box with two compartments. One was a square compartment without a lid, serving as the light side (40 × 40 cm), while the other was a smaller rectangular compartment with a lid, creating the dark side (20 × 40 cm). These two compartments were connected by a 5 × 5 cm door, and the entire apparatus had a height of 30 cm. The center of the lit compartment maintained a stable luminosity of 400 lux, while the dark compartment remained without any light source. Mice were introduced into the apparatus in the light side, facing the door, and allowed to explore for a duration of 5 min. The mice's movements were tracked and recorded using the ANY-maze software. In this test, anxiety-like behavior was evaluated based on the time spent in the dark compartment.

### Marble burying test

The experiment was conducted following the methodology described (Deacon, 2006; Kedia and Chattarji, 2014). The apparatus used for the test consisted of an open, transparent plastic box measuring 40 × 25 × 20 cm, filled with bedding that had a depth of 6 cm. The test involved placing 20 dark marbles, each 16 mm in diameter, spaced evenly in a 4 × 5 grid on the surface of the bedding. Mice were placed in the middle of the box and were free to

explore for 20 min under 300 lux. Upon completion of the experiment, the mice were removed from the box, and the number of buried marbles was visually assessed. A marble was considered "buried" when more than two-thirds of its surface was covered by the bedding. The evaluation of an elevated number of buried marbles is indicative of an anxious profile in the mice.

### Social confrontation tube test

The social confrontation tube test (SCTT) was conducted on mice cohabitating for a duration of 5 weeks as previously described (Larrieu et al, 2017). Each mouse underwent training to cross a clear Plexiglas tube (diameter: 3 cm; length: 30 cm) five times from each end over two consecutive days. The tube's diameter allowed unidirectional movement for adult mice. During the habituation phase, retreat or cessation of movement prompted gentle encouragement by touching the tail with a plastic stick. Between trials, the tube was cleaned with a 70% ethanol solution to eliminate smell, urine, or feces. Following the two-day habituation, social ranks were assessed over 9 consecutive days. Before the confrontation phase, each mouse was retrained to cross the tube from each end. Using a round-robin design, pairwise confrontations were conducted within social groups, resulting in six trials per cage of four mice. Two mice were simultaneously guided by the tail at the opposite tube ends until they reached the middle of the tube, and the time spent in the tube was recorded until one mouse compelled its cage mate to retreat. The mouse that retreated was identified as the 'loser' for that specific trial. Subsequently, for each trial, the same mouse alternated being placed in the tube from each end. Social dominance was quantified by calculating the percentage of winning time, and mice were ranked from 1 to 4, with Ranks 1 and 2 signifying the most dominant mice and Ranks 3 and 4 indicating the most subordinate mice.

The SCTT was also used to evaluate situational social dominance (as opposed to hierarchical dominance). Mice from each treatment group (TMZ or NaCl) were paired with mice from the other group, attempting to match body weight and trait anxiety. The couples were tested twice a day, in the morning and afternoon. To prevent potential confounding effects and biases, we took precautions to avoid pairing innate dominant saline mice with innate subordinate TMZ mice. Following an initial random pairing, the outcome yielded dominant mice from both the TMZ and saline groups. In a subsequent pairing, we matched dominant TMZ mice with dominant saline mice that never met before with similar body weight and trait anxiety. The second round of confrontation revealed the social dominance of mice based on the number of wins across the 5 days. Mice with a percentage of victories higher than 50% were considered dominant. The Plexiglas tube was cleaned between each trial with 70% ethanol to remove odors, urine, and feces.

### Anxiety score

The anxiety scores were a simplified version of principal component-based analyses (Harrison et al, 2020; Heinz et al, 2021) and encompassed several anxiety tests to obtain a general profile of anxiety, as previously described (Bosch-Bouju et al, 2016; Cherix et al, 2020; Di Miceli et al, 2022; Larrieu et al, 2017). The scores were derived from the normalization of the values for the combination of individual anxiety tests (time spent in the dark chamber during LDT, time spent into the closed arm in an EPM,

time spent in thigmotaxis in an OFT and marbles buried). The normalization involved adjusting each animal's value by subtracting the minimum value of the entire population and then dividing this result by the difference between the maximum and minimum values of the entire population: (x − min value)/(max value − min value). This method generates scores distributed on a scale from 0 to 1, with a score of 1 indicating high anxiety. The rationale for employing different combinations of tests is to increase the robustness of our findings by exploring whether TMZ would consistently influence anxiety-like behavior across a broader range of validated paradigms. While the open field, light-dark, and elevated plus maze tests are all well-established tools for assessing anxiety, we included additional measures, such as the marble burying test in the second experiment, to account for potential variations in anxiety responses that different tasks might detect. This approach also allowed us to confirm the reliability of TMZ's effect by showing that any observed behavioral changes were not limited to a specific subset of tests but rather generalizable across multiple anxiety paradigms. Normalizing the results across three tests in each experiment ensures consistency in our scoring method, even as we explore diverse anxiety tests. The variation of the test combinations enhances the rigor of the study without compromising the validity of the anxiety score.

## Histology

After all the behavioral experiments the mice were sacrificed with a lethal injection of pentobarbital (10 mL/kg, Sigma-Aldrich, Switzerland) and transcardially perfused with saline solution (NaCl 0.9%) followed by 4% paraformaldehyde (PFA) solution (Sigma-Aldrich, Switzerland). Brains were removed and postfixed with PFA 4% at 4 °C overnight. Then brains were transferred in a 30% sucrose solution for 3 days before being frozen at −20 °C until slicing. Coronal frozen sections of a thickness of 40 μm. were sliced with a microtome or cryostat (microtome, Leica Microsystems; cryostat, Leica CM3050S) to obtain hippocampal sections conserved in a cryoprotectant solution (30% glycerol, 30% ethylene glycol and 40% PBS 1 M) at −20 °C until immunofluorescence staining.

## Immunofluorescence staining

One out of 6 slices containing hippocampal tissue were chosen to cover the whole dentate gyrus and used for immunostaining. For detection of BrdU-, IdU- and CldU-positive cells, immunohistochemistry was performed using a formic acid pre-treatment (formamide 50%, 10% SSC 20X and 40% MilliQ water) at 60 °C for 2 h followed by DNA denaturation in 2 M HCl for 30 min at 37 °C and rinsed in 0.1 M borate buffer pH 8.5 for 15 min and 6 times in PBS 0.1 M for 10 min. The slices were incubated in blocking solution (0.3% TritonX-100 and 10% horse serum in PBS) at room temperature for 1 h and then incubated under agitation at 4 °C overnight in a blocking solution containing the following primary antibodies: Rat anti-BrdU/CldU (Abcam, 1:1000, ab6326), and Mouse anti-IdU monoclonal antibody (Abcam, 1:500, ab181664). After being washed again in PBS, sections were incubated for 2 h in either of the following secondary antibodies: Alexa Fluor 488 goat anti-mouse (Invitrogen, 1:300, A11032), and Alexa Fluor 594 goat anti-rat (Invitrogen, 1:300, A11007). Sections were finally incubated for 10 min in 4′,6-diamidino-2-phenylindole

(DAPI, 2 µg/ml in PBS) solution to mark the cell nuclei. Sections were mounted onto glass slides and cover-slipped using FluorSave (Millipore). Imaging was performed using a Nikon NI-E Spinning disk microscope. BrdU-, CldU- and IdU-positive cells were counted using the NIS elements Nikon software (ver 6.02) and automatized supervised counting. The experimenter was blind to the identity of the mice. The density of BrdU-, CldU- and IdU-positive cells was quantified by dividing the number of BrdU+ cells by the corresponding volume of DG expressed in mm$^3$.

## Statistical analyses

Statistical analyses were carried out with Prism 9 (GraphPad Software, San Diego, CA 92108, USA), using an alpha level of 0.05. We eliminated outlier values that deviated more than 2 standard deviations from the average. All data are presented as mean ± SEM. Data were tested for normality using the Shapiro-Wilk test. For normally distributed measures, we used an unpaired, two-tailed Student t-test to estimate differences between the two groups in trait anxiety (EPM, OFT, LDT, MBT), in body weight, in social dominance (SCTT) and in hippocampal neurogenesis (CldU-, IdU- and BrdU-immunolabeled cells). To evaluate the effect of TMZ on social hierarchy establishment and situational social dominance a Chi-square test has been performed. The social-confrontation tube test results were obtained by using either an unpaired, two-tailed Student t-test to estimate differences between the two groups or a two-way analysis of variance ANOVA with repeated measure, with treatment and days as fixed factors. Analyses were followed by a Bonferroni post-hoc test when it was appropriated. To correlate the cell survival with the anxiety scores and percentage of winning, a Pearson linear regression analysis was carried out. To test the accuracy of the model with adolescent neurogenesis and anxiety as predictors, we plotted the true positive values (Specificity) against the false positive values (Sensitivity) using the Receiver Operating Characteristic (ROC) curve.

## Data availability

All data presented in this manuscript has been deposited on Zenodo: https://doi.org/10.5281/zenodo.13821490.

The source data of this paper are collected in the following database record: biostudies:S-SCDT-10_1038-S44319-025-00367-y.

## Peer review information

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

## Acknowledgements

The authors wish to thank the confocal imaging facility of the University of Lausanne (CIF) for access and training to confocal microscopy, Benjamin Boury-Jamot, Fulvio Magara and the Centre d'études du Comportement of the Center for Psychiatric Neurosciences for advice and help with behavioral studies. This project was funded by the Swiss National Science Foundation (Grant Number 310030_201015).

## Author contributions

**Fabio Grieco**: Formal analysis; Supervision; Investigation; Methodology; Writing—original draft; Writing—review and editing. **Atik Balla**: Investigation. **Thomas Larrieu**: Conceptualization; Data curation; Formal analysis; Supervision; Validation; Investigation; Methodology; Writing—original draft; Writing—review and editing. **Nicolas Toni**: Conceptualization; Data curation; Supervision; Funding acquisition; Validation; Writing—original draft; Project administration; Writing—review and editing.

Source data underlying figure panels in this paper may have individual authorship assigned. Where available, figure panel/source data authorship is listed in the following database record: biostudies:S-SCDT-10_1038-S44319-025-00367-y.

## Disclosure and competing interests statement

The authors declare no competing interests.

# Expanded View Figures

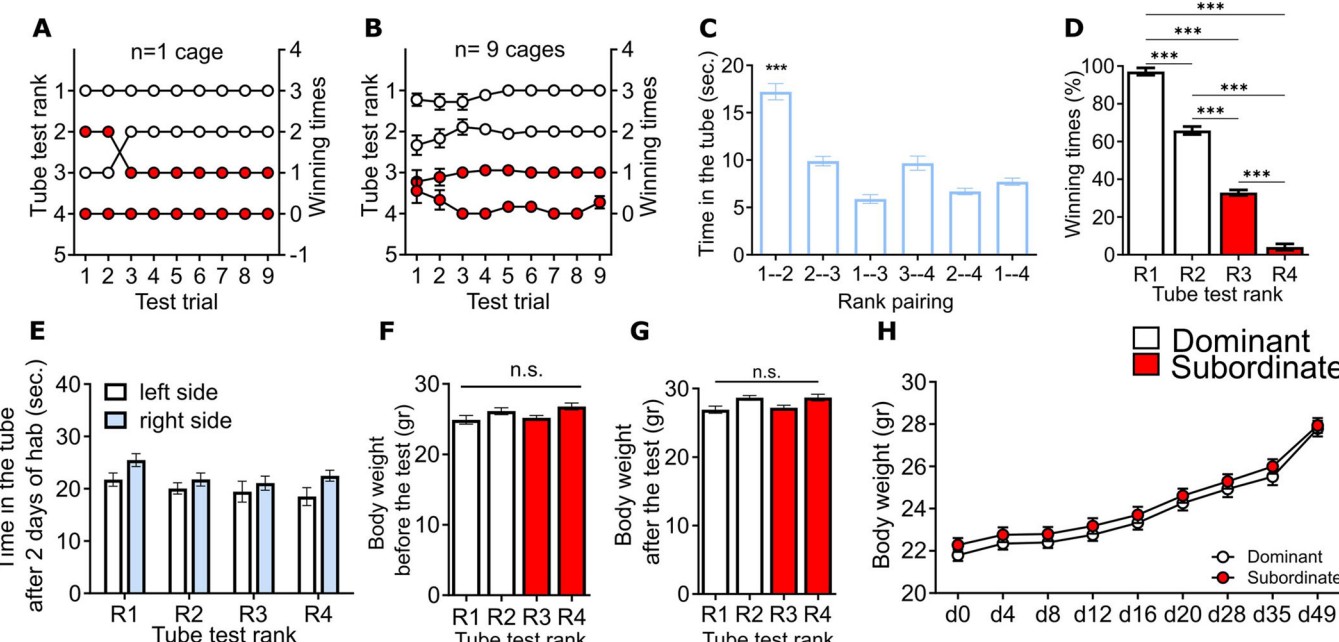

**Figure EV1. Hierarchical rank assessed via a social confrontation tube test.**

(A) Example of tube test ranks and winning times over trials for one cage ($n = 1$ cage). (B) Average ranks across nine cages over 9-day trials ($n = 9$ cages). (C) Time spent in the tube by rank pairings ($F_{(5,48)} = 49.69$, $p < 0.001$, one-way ANOVA, $n = 9$ cages per pairing). (D) Winning percentages by rank after 9 days of confrontations ($F_{(3,42)} = 494.6$, $p < 0.0001$, one-way ANOVA, $n = 9$ cages per pairing). (E) Average time spent in the tube during the 2-day habituation phase by final rank (interaction: $F_{(3,64)} = 0.41$, $p = 0.7442$; rank effect: $F_{(3,64)} = 2.44$, $p = 0.0719$; side effect: $F_{(1,64)} = 7.787$, $p = 0.0983$; two-way ANOVA, $n = 9$ cages/group). (F, G) Body weight by rank before (F) and after (G) the tube test (before: $t34 = 0.89$, $p = 0.376$, unpaired t-test R1-R2 vs. R3-R4, two-tailed, $n = 18$ mice/group; after: $t34 = 0.30$, $p = 0.761$, unpaired t-test R1-R2 vs. R3-R4, two-tailed, $n = 18$ mice/group). (H) Body weight evolution in dominant versus subordinate mice. Histograms show average ± SEM; ***$p < 0.001$; ns = not significant. Source data are available online for this figure.

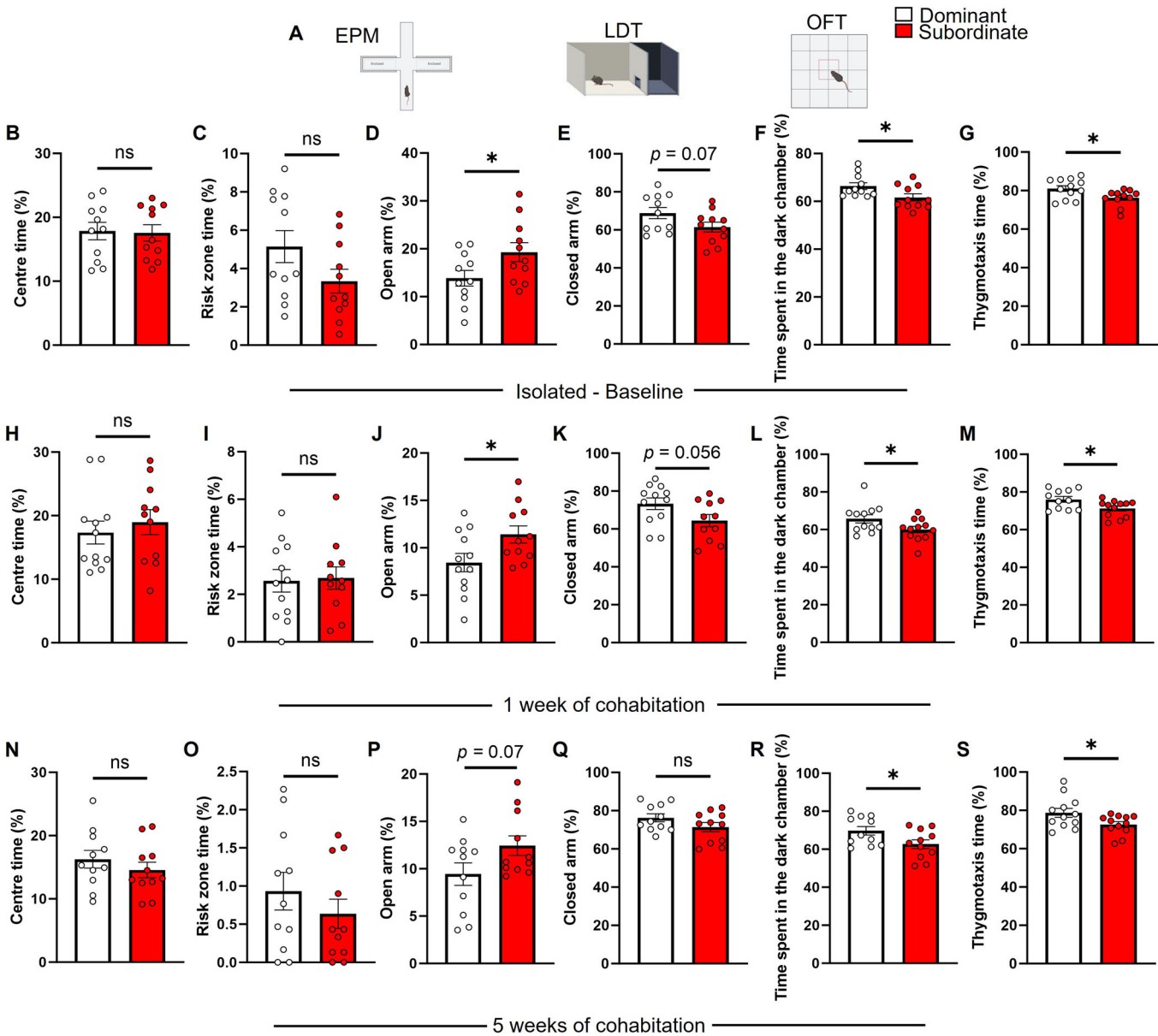

**Figure EV2. Anxiety measures in dominant and subordinate individuals.**

(A) Diagram of anxiety tests conducted. (B–E) Baseline behaviors in an EPM expressed as a percentage of time spent in the: (B) Center zone (t20 = 0.156, p = 0.823; unpaired t-test, two-tailed; n = 11/group). (C) Risk zone (t20 = 1.733, p = 0.098; unpaired t-test, two-tailed; n = 11/group). (D) Open arm (t20 = 2.123, p = 0.046; unpaired t-test, two-tailed; n = 11/group). (E) Closed arm (t20 = 1.890, p = 0.073; unpaired t-test, two-tailed; n = 11/group). (F) Percentage of time spent in the dark chamber in a LDT (t20 = 2.29, p = 0.033; unpaired t-test, two-tailed; n = 11/group). (G) Percentage of time in thigmotaxis in an OFT (t20 = 2.44, p = 0.023; unpaired t-test, two-tailed; n = 12 dominants, n = 11 subordinates). (H–K) Behavior in an EPM after one week of cohabitation expressed as a percentage of time spent in the: (H) Center zone (t21 = 0.618, p = 0.543; unpaired t-test, two-tailed; n = 12 dominants, n = 11 subordinates). (I) Risk zone (t21 = 0.177, p = 0.861; unpaired t-test, two-tailed; n = 12/group). (J) Open arm (t21 = 2.231, p = 0.037; unpaired t-test, two-tailed; n = 12/group). (K) Closed arm (t21 = 2.021, p = 0.056; unpaired t-test, two-tailed; n = 12/group). (L) Percentage of time spent in the dark chamber in a LDT after one week of cohabitation (t22 = 2.093, p = 0.048; unpaired t-test, two-tailed; n = 12/group). (M) Percentage of time in thigmotaxis in an OFT after one week of cohabitation (t21 = 2.391, p = 0.026; unpaired t-test, two-tailed; n = 11 dominants, n = 12 subordinates). (N–Q) Behavior in an EPM after one week of cohabitation expressed as a percentage of time spent in the: (N) Center zone (t20 = 0.911, p = 0.373; unpaired t-test, two-tailed; n = 11/group). (O) Risk zone (t20 = 0.949, p = 0.354; unpaired t-test, two-tailed; n = 11/group). (P) Open arm (t20 = 1.901, p = 0.072; unpaired t-test, two-tailed; n = 11/group). (Q) Closed arm (t20 = 1.545, p = 0.138; unpaired t-test, two-tailed; n = 11/group). (R) Percentage of time spent in the dark chamber in a LDT after five weeks of cohabitation (t20 = 2.164, p = 0.043; unpaired t-test, two-tailed; n = 11/group). (S) Percentage of time in thigmotaxis in an OFT after five weeks of cohabitation (t22 = 2.188, p = 0.04; unpaired t-test, two-tailed; n = 12/group). Histograms show average ± SEM. *p < 0.05; ns = not significant; n = number of animals. Source data are available online for this figure.

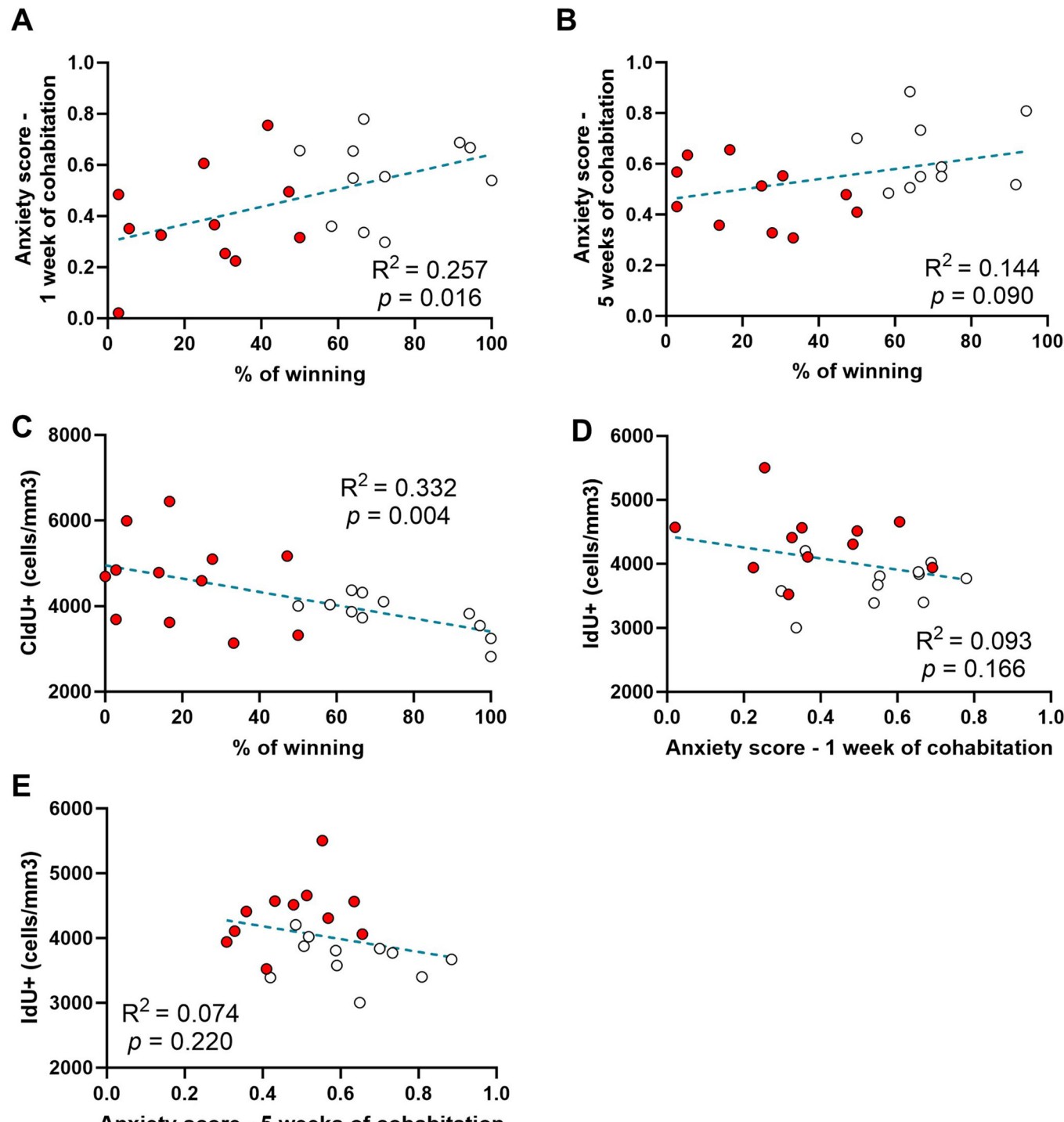

**Figure EV3. Correlations between AHN, anxiety, and social dominance.**

(A, B) Correlations between anxiety and percentage of winning during the SCTT after one week of cohabitation (A) and five weeks of cohabitation (B). (A, $R^2 = 0.257$, $p = 0.016$, simple linear regression, $n = 11$ mice per group. B, $R^2 = 0.144$, $p = 0.090$, simple linear regression, $n = 11$ subordinates, $n = 10$ dominants). (C) Correlation between the number of CldU-positive cells and percentage of winning during the SCTT ($R^2 = 0.332$, $p = 0.004$, simple linear regression, $n = 12$ subordinates, $n = 11$ dominants). (D, E) Correlation between the number of IdU-positive cells and the anxiety level after one week (D) and five weeks of cohabitation (E) (D, $R^2 = 0.093$, $p = 0.166$, simple linear regression $n = 11$ subordinates, $n = 10$ dominants. E, $R^2 = 0.074$, $p = 0.220$, simple linear regression, $n = 11$ mice per group). Blue line shows the linear regression. Source data are available online for this figure.

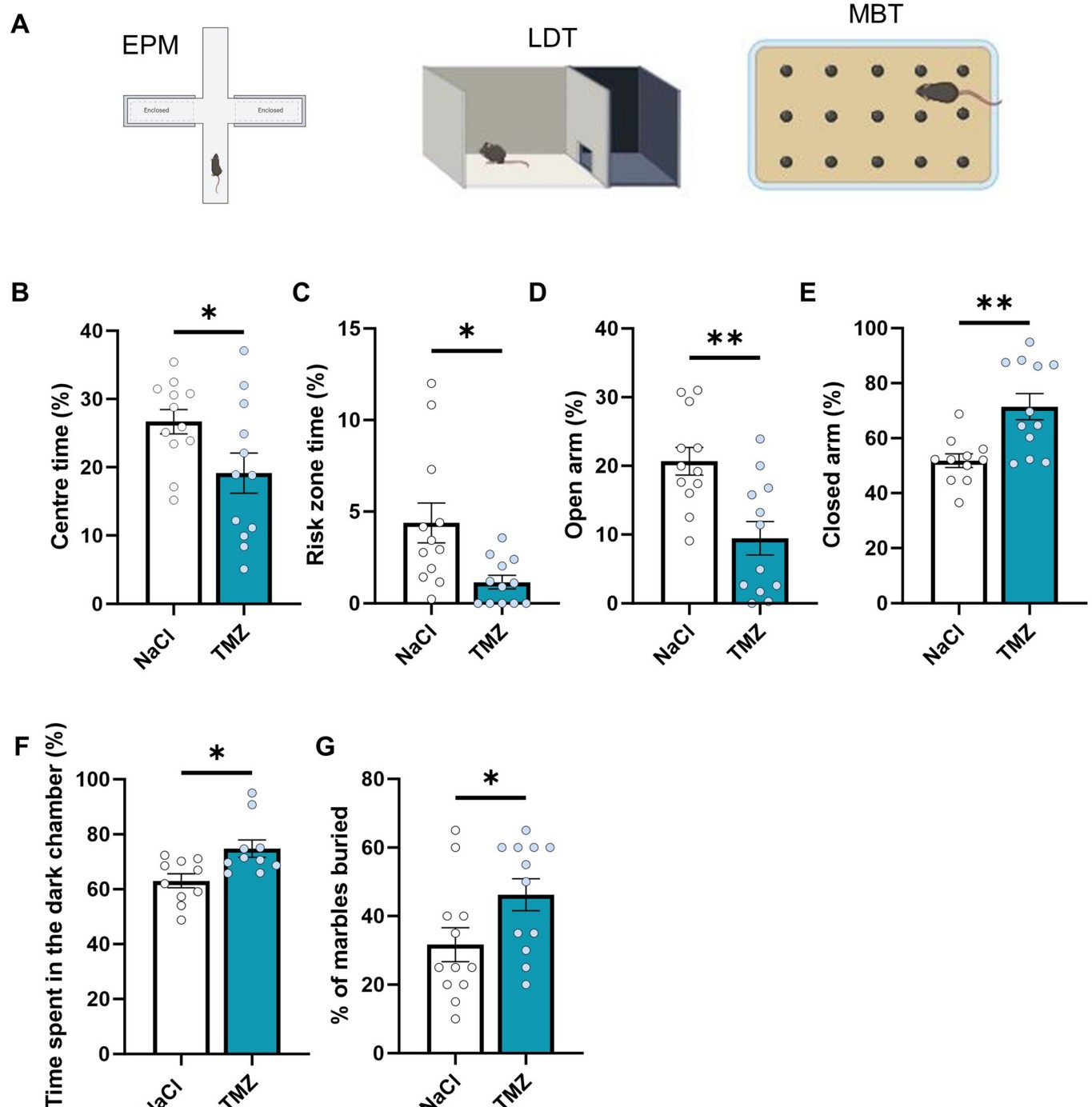

**Figure EV4. Effect of TMZ on anxiety after group formation.**

(A) Graphic representation of the behavioral tests used to assess anxiety. (B–E) Evaluation of several behaviors during EPM: percentage of time spent in the center (B), in risk zone (C), in the open arm (D) and in closed arm (E) (B, t22 = 2.193, p = 0.039, unpaired t test, two-tailed, n = 12 mice per group. C, t22 = 2.813, p = 0.0101, unpaired t test, two-tailed, n = 12 mice per group. D, t22 = 3.556, p = 0.0018, unpaired t test, two-tailed, n = 12 per group. E, t21 = 3.553, p = 0.0019, unpaired t test, two-tailed, n = 12 mice per TMZ, n = 11 per NaCl). Percentage of time in the dark chamber in LDT (F) and of marbles buried in MBT (G) (F, t18 = 2.87, p = 0.01, unpaired t test, two-tailed, n = 10 mice per group. (G), t22 = 2.15, p = 0.042, unpaired t test, two-tailed, n = 12 mice per group). Histograms show average ± SEM, *p < 0.05, **p < 0.01. Source data are available online for this figure.

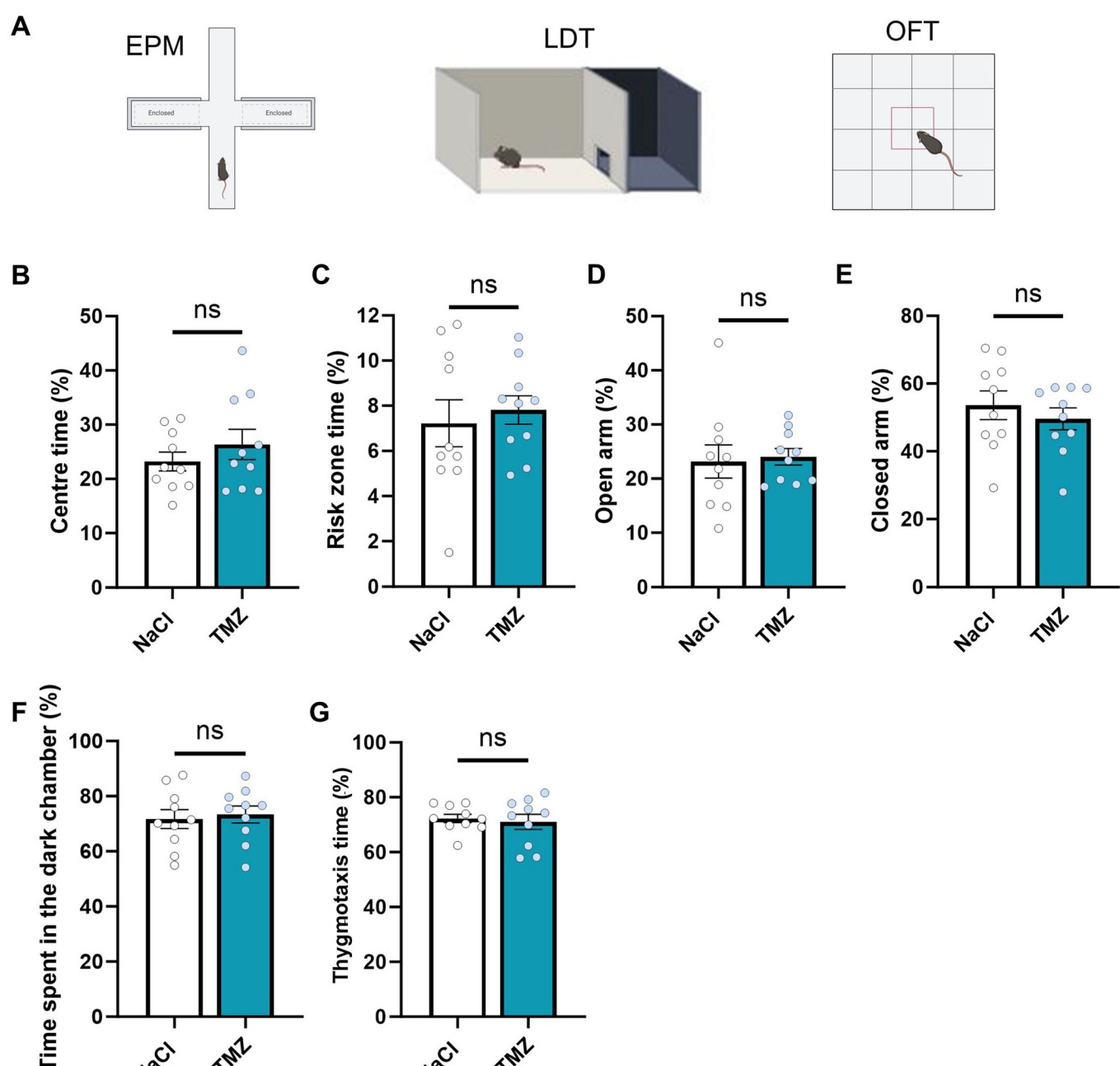

**Figure EV5. Effect of TMZ on anxiety before group formation.**

(A) Graphic representation of behavioral tests used to assess anxiety. (B–E) Evaluation of several behaviors during EPM: percentage of time spent in the center (B), in risk zone (C), in the open arm (D) and in closed arm (E) (B, t18 = 0.956, p = 0.3514, unpaired t test, two-tailed, n = 10 mice per group. C, t18 = 0.485, p = 0.6334, unpaired t test, two-tailed, n = 10 mice per group. D, t18 = 0.259, p = 0.7986, unpaired t test, two-tailed, n = 10 mice per group. E, t18 = 0.757, p = 0.4587, unpaired t test, two-tailed, n = 10 mice per group). Percentage of time spent in the, in the dark chamber in LDT (C) and in thygmotaxis during OFT (D) (B, t18 = 0.75, p = 0.458, unpaired t test, two-tailed, n = 10 mice per group; C, t18 = 0.36, p = 0.720, unpaired t test, two-tailed, n = 10 mice per group; D, t18 = 0.40, p = 0.690, unpaired t test, two-tailed, n = 10 mice per group). Histograms show average ± SEM, ns = not significant. Source data are available online for this figure.

