## [Peer Review File · EMBO Reports]

Natural variations of adolescent neurogenesis and anxiety predict the hierarchical status of adult inbred mice

Fabio Grieco, Atik Balla, Thomas Larrieu, and Nicolas Toni

Corresponding authors: Nicolas Toni (nicolas.toni@unil.ch), Thomas Larrieu (thomas.larrieu@chuv.ch)

Review Timeline:

Submission Date:	13th Mar 24
Editorial Decision:	23rd Apr 24
Revision Received:	21st Sep 24
Editorial Decision:	11th Nov 24
Revision Received:	20th Dec 24
Accepted:	7th Jan 25

Editor: Esther Schnapp

Transaction Report:

Dear Dr. Toni,

Thank you for the submission of your manuscript to EMBO reports. We have now received the full set of referee reports as well as referee cross-comments that are all pasted below.

As you will see, the referees acknowledge that the findings are potentially interesting. However, they also raise several and important concerns that would need to be addressed in order to proceed with your manuscript here. Especially referee 1 has several concerns, and I asked referees 2 and 3 whether they agree with these, and they do but also feel that you should be given a chance to revise your study and address all concerns. This is a borderline case, but if you think that you can address all comments, I am happy to invite you to revise your manuscript. Please let me know if you have any questions or comments and we can discuss the exact revision requirements further, also in a video chat, if you like.

I would thus like to invite you to revise your manuscript with the understanding that the referee concerns must be fully addressed and their suggestions taken on board. Please address all referee concerns in a complete point-by-point response. Acceptance of the manuscript will depend on a positive outcome of a second round of review. It is EMBO reports policy to allow a single round of major revision only and acceptance or rejection of the manuscript will therefore depend on the completeness of your responses included in the next, final version of the manuscript.

We realize that it is difficult to revise to a specific deadline. In the interest of protecting the conceptual advance provided by the work, we recommend a revision within 3 months (24th Jul 2024). Please discuss the revision progress ahead of this time with the editor if you require more time to complete the revisions.

- 1) A data availability section providing access to data deposited in public databases is missing. If you have not deposited any data, please add a sentence to the data availability section that explains that.
- 2) Your manuscript contains statistics and error bars based on $n=2$. Please use scatter blots in these cases. No statistics should be calculated if $n=2$.

5) a complete author checklist, which you can download from our author guidelines

<<https://www.embopress.org/page/journal/14693178/authorguide>>. Please insert information in the checklist that is also reflected in the manuscript. The completed author checklist will also be part of the RPF.

6) Please note that all corresponding authors are required to supply an ORCID ID for their name upon submission of a revised manuscript (<<https://orcid.org/>>). Please find instructions on how to link your ORCID ID to your account in our manuscript tracking system in our Author guidelines

<<https://www.embopress.org/page/journal/14693178/authorguide#authorshipguidelines>>

10) Regarding data quantification (see Figure Legends:

<https://www.embopress.org/page/journal/14693178/authorguide#figureformat>)

- the name of the statistical test used to generate error bars and P values,

- the number (n) of independent experiments (please specify technical or biological replicates) underlying each data point,

- the nature of the bars and error bars (s.d., s.e.m.),

- If the data are obtained from n Program fragment delivered error ``Can't locate object method "less" via package "than" (perhaps you forgot to load "than"?) at //ejpvfs23/sites23b/embor_www/letters/embor_decision_revise_and_review.txt line 56.' 2, use scatter blots showing the individual data points.

You are able to opt out of this by letting the editorial office know (emboreports@embo.org). If you do opt out, the Review Process File link will point to the following statement: "No Review Process File is available with this article, as the authors have

chosen not to make the review process public in this case."

I look forward to seeing a revised form of your manuscript when it is ready.

Referee #1:

The work of Grieco and coworkers is focused on the causal relationship between social dominance, anxiety, stress resilience and neurogenesis. Although a number of papers have been published on the association of dominance with anxiety/stress vulnerability, as well as association between anxiety and neurogenesis, the current manuscript addressed the relationship between dominance and neurogenesis. Based on experiments in the first part of the manuscript, the authors claim that dominant individuals have lower adult neurogenesis than submissive individuals. The second part of the work tests whether reducing neurogenesis increases dominance status. Although the idea behind the possible link between dominance rank and neurogenesis and vice versa is novel and interesting, the data do not seem to support this connection. There are also technical issues with the neurogenesis manipulation experiments.

Major points.

1. It is stated (line 106-108) that dominant individuals have lower BRDU+ cell counts in the dorsal DG (i.e., reduced neurogenesis), but corresponding Fig. 2G and corresponding legend do not show a neurogenesis difference between Control dominant and submissive mice (there is no posthoc test that would indicate a difference). Further, even the rank effect is trend only (indicated by a * that usually signals significance). In any event, the rank effect includes both Control and Stress mice, but the important groups to support the claim are the Controls because they are used to justify the neurogenesis phenocopy experiment.
2. Fig 2H. Same issue with DCX labeling intensity, except that rank effect (for Control and Stress) is significant; but again, the important comparison would be between dominant and submissive Controls by a posthoc analysis, which is not indicated in the Fig./legend.
3. Since neurogenesis between dominant and submissive Controls does not seem to be different, justification to manipulate neurogenesis by TMZ is unfounded. Beyond this problem, TMZ causes a massive reduction, around 50% in neurogenesis, while the "non-significant" difference between Control dominant and submissive is around 15% (judging from the figure). Therefore, the TMZ experiment does not really phenocopy the dominance associated (but not significant) reduction in neurogenesis in Control mice.
4. The experimental design of the TMZ experiment is not optimal. In a proper design that tests causality between neurogenesis and social status, subordinates are selected for TMZ to see if chemically reducing neurogenesis converts these animals dominant. Dominant mice should not respond to TMZ as they cannot be more dominant.
5. Inhibition of neurogenesis by TMZ is unlikely to be limited to the dorsal DG, undercutting the association between dorsal DG neurogenesis and dominance. There are region specific manipulations of neurogenesis that are superior to chemical ablation.

Additional comments

1. The introduction is a little rambling and difficult to follow. Anxiety is linked to dominance but then to subordination. Perhaps the authors wanted to say that the link is tenuous and both increased and reduced anxiety and stress susceptibility have been linked to dominance status. It would be useful to reorganize the intro and outline a clear rationale for the studies. Introduction of stress is particularly confusing as the paper is not about stress but rather about the relationship between social rank and neurogenesis. The purpose of Stress mice in this context is unclear. Also, this makes the first and second parts of the manuscript disjointed.
2. Why are BRDU numbers are per slice in Fig. 1 while in the TMZ experiment in Fig. 2 the numbers are normalized to volume. The latter would be much better as mismatch of sections across animals and groups can introduce confounds.
3. Full statistical analyses should be presented with posthoc data in the fig legends. This would help to understand whether changes are significant or not.
4. It seems that BRDU positive cells in the ventral DG is higher in Control dominant vs. submissive mice (Fig. 2C), which is unusual as it would be the opposite to the claimed difference in the dorsal DG. Moreover, this may contribute to the CRS induced reduction in neurogenesis in dominant but not submissive mice.
5. Fig 2I. Correlation is shown for BRDU. What about DCX? If no correlation, why?
6. Calculation of anxiety score is not clear. It is mentioned that normalized values are used from 3 anxiety tests (time spent in dark/closed ...). Does this mean that dark/closed time was used or that light/open time was normalized to dark/closed? Was this approach of calculating anxiety validated in a well-established anxiety model? I found no reference.
7. The authors claim that dominance is established gradually once animals are cohoused from 6 weeks of age. I assume that mice were cohoused with others from weaning to 6 weeks of age that could be sufficient to develop the dominant and

submissive phenotypes. Dominant mice likely continue to be dominant in a new group. This is supported by published data showing that winners keep winning. Do the authors have evidence that experience prior cohousing at 6 weeks of age has no effect on dominance?

8. The number of animals in each group should be specified.

9. Label for the dominant and submissive groups should be in Fig2, like in Fig 1.

Referee #2:

Summary:

Hierarchical dominance behavior in mice can be associated with higher trait anxiety and emerges in response to different stressors. Here the authors set out to explore how adult hippocampal neurogenesis may mediate the interplay between anxiety levels and dominance behavior in mice. The authors used SCTT to establish dominance and assessed neurogenesis levels and did functional manipulations of neurogenesis levels. Their results show that dominant mice display higher anxiety scores and vulnerability to chronic restraint stress, as well as impaired adult neurogenesis compared to subordinate mice after CRS. Adult neurogenesis depletion increased hierarchical dominance as well as situational dominance, irrespective of anxiety scores, indicating that neurogenesis can modulate dominance behavior independently of anxiety levels.

Major points:

- Rationale for the putative role of adult hippocampal neurogenesis in the establishment of dominance behavior could be clearer/more developed.
- The authors use an anxiety score which considers the animals' behavior across different anxiety assays. The behavioral measures assessed in each of these assays should be included in supplemental information. Additionally, while this score can provide an account of the underlying anxiety of the animal, an explanation of the tests chosen and of the measures used for each test should be included. Namely, why use measures of time spent in the "safe" feature of the apparatus for each paradigm (closed arm in EPM, dark chamber in LDT, walls in the OFT/NO) as opposed to time engaging in exploratory behavior in the more anxiogenic areas of the respective tests. Finally, please clarify why are different combinations of tests used in each experiment to generate an anxiety score.
- To claim that increased anxiety is a pre-existing trait in mice that become dominant, would require testing anxiety at baseline (e.g. single housing before cohousing, and testing anxiety levels before cohousing), and then observe if a priori innate anxiety predicts dominance behavior, and whether this is or not neurogenesis dependent. As is, the authors observed consistent higher anxiety levels once the animals had already been cohoused 1 or several weeks and established dominance, but not a priori trait anxiety.
- Regarding the hypothesis that a priori levels of neurogenesis influence dominance behavior: this is not clearly addressed with this experimental design, as the timing of BRDU injections is always after cohousing and manipulations have started and may be influencing the levels of new neurons labelled and their survival and maturation. Without live imaging throughout the timeline, it is hard to make claims about a priori levels of neurogenesis. The discussion should clarify this.
- To manipulate neurogenesis, the authors used TMZ, which affects olfactory bulb neurogenesis as well as hippocampal neurogenesis, as if referred in the discussion. It would be crucial, for the experiments that utilize this treatment, to assess the effect of TMZ in odor discrimination and in social recognition.
- Since the hippocampus is involved in social memory and social recognition, namely through its CA2 subregion, it would be important to also control the effects of manipulations of hippocampal neurogenesis in a task of social memory, to see if disrupted social memory (recognition and discrimination) could explain the effect of neurogenesis in dominance behavior.

Minor points:

Introduction: a review article (reference 1) is used multiple times instead of the specific relevant research articles.

Figure 2:

- For DCX, since the measure used was intensity instead of cell counts, elaborate on how intensity was measured and normalized.
- Here the correlation analysis was only shown for mice post CRS. The authors could also show analysis in the control animals, even if only in supplement; likewise for analysis of DCX levels vs. anxiety scores.
- To further corroborate their point, using the data already collected, the authors could analyze the correlation between anxiety scores and rank level, and between neurogenesis levels and rank level; or compare the animals that rank 1st vs animals that rank 4th.

Figure 4:

- The experimental design is not fully clear. Please clarify if the data displayed are for the first and second or only the second round of tests, after determining "innate" dominance of mice in both control and TMZ groups and then making new pairs? How were the animals matched for innate anxiety if testing of anxiety was done after the SCTT - was there a previous batch of anxiety tests done before the second round of testing? Please elaborate on the timeline of this experiment.

Typos:

Line 61: "an observational study [and] two interventional studies"

Line 129: "SCTT test" -> "SCTT"

Line 212: ablation or reduction?

Line 282: "in human[s]"

Line 488: "by using the number of [DAPI or NeuN] positive cells"?

In conclusion:

This manuscript presents very interesting new data which shows that dominant mice display higher levels of anxiety, and that neurogenesis levels differ between dominant and submissive mice (in the dDG) and reflect their different susceptibilities to stress (in the vDG). Additionally, manipulations of neurogenesis in a non-targeted way affected dominance behavior independently of anxiety levels, but this effect remains to be explained. To support its main claims further justification and additional experiments are required. I advise major revisions requiring re-review.

Referee #3:

The authors examined the role of adult neurogenesis in linking social rank and anxiety level. This is a very interesting study, but the way it is presented does not render justice to the very nice data of this work. Indeed adult neurogenesis has been shown to "regulate" social hierarchy in one hand and anxiety-like behaviour in another one. In the present paper, both traits were examined in the same animals in basal condition and in response to chronic stress. This is a novel aspect of the story that is underexploited in the manuscript. Just to give an example: the title : "Adult neurogenesis regulates social dominance and anxiety". The fact that adult neurogenesis regulates these traits is already known.

The way the manuscript is written is complex and often does not correspond exactly to the experiments carried out. For example (L60), the authors write that they will investigate "whether inter-individual differences in adult neurogenesis may underlie differences in anxiety and dominance behavior". The design of the experiment (between-group analysis rather than a within-group analysis) does not allow to study inter-individual differences per se.

It's also hard to know what message to retain. For example (L22) the authors wrote "these results indicate that adult neurogenesis regulates hierarchical and situational dominance behavior along with anxiety-related behavior, independently from each other. This conclusion is unclear as in figure 3 it is shown that reduction of cell proliferation is impacting both traits. In Figure 4, animals are extremely anxious which explain the lack of effects on anxiety.

METHODS

- The experimental design should be described after the description of the tasks!
- In this study, adolescent animals (7 week at the beginning of testing) were used. It is understandable that working with adult male mice (10-12 weeks) is complicated by their aggressive behaviour. However, it would be appropriate to state in the text why such young animals were chosen.
- L318 Add the number of animals per experiment and per group. Are the number of animals similar in Fig1C left and right panel and in Fig1 D,E
- L467 The authors wrote "chosen at regular intervals ". Please indicate the sampling probability (1 in X).
- L485 why the number of Dcx cells were not counted manually? What does mean DCX labelling intensity ? This approach can lead to misleading results.
- L487: It is indicated that the volume density has been measured. However, the results are expressed as nb of cell/section. The quantification of cells (nb of cell/section) is unconventional in the field (considered as biased quantification). Therefore, I would recommend to present the results in volume density or to use the optical fractionator, as the number of cells was evaluated along the septotemporal axis (Nb of cells = sum of nb of cells/section X the inverse of the sampling probability).
- L322: EPM, the OF/NO test, and the LDT were measured in experiment 1. The results for each task should be detailed in addition to the anxiety score. Were mice hyperactive? Note that on the figure 1, EPM has been forgotten!
- L324 How the mice were separated into two groups? Were the anxiety score of the control and the stress group similar before CRS?
- L329 The marble burying test (MBT) is measuring anxiety

RESULTS

- L78: results of the plus maze are lacking
- L99: adult neurogenesis has been shown to be involved in anxiety-like behaviour (Revest et al.2009). Note that the authors confirmed this conclusion using TMZ.
- L94: The authors wrote: "we observed that after CRS, dominant mice displayed decreased stress resilience compared to

subordinate mice. This was reflected by an increased number of buried marbles in the marble burying test (Fig. 1D) and the development of social avoidance toward an unfamiliar CD1 mouse in a social interaction test (SI) (Fig. 1E). In contrast, subordinate mice responded similarly to unstressed mice in both tests, suggesting a resilient phenotype against chronic restraint stress (Fig. 1D, E). It is a convoluted way of saying that CRS causes further anxiety in the dominant but not in the subordinate.

- L118 Inhibiting adult neurogenesis increases social dominance behavior and anxiety
- Are anxiety score before and after CR correlated (within each group)
- Are anxiety score before CR predictive of the level of adult neurogenesis (within each group)
- The effects of CRS on adult neurogenesis are less obvious. Isn't this due to the housing conditions: stressed animals are housed with control animals (a procedure not typically used in behavioural studies) which could buffer the effects of stress. This needs to be discussed.
- Figure 3 and 4 : please provide individual performances in the EPM, the OF/NO test, and the LDT
- Figure 4: to which extent adult neurogenesis (Dcx) was reduced?

DISCUSSION

• LN 173. The authors wrote we examined the role of adult hippocampal neurogenesis in shaping social dominance behavior. This would have been the case if neurogenesis was measured before measuring social dominance behavior. Rather their findings revealed that the level of anxiety predicts social rank that influence vulnerability to CRS: subordinates are more resilient to stress compare to dominants.

•Line 187 "but is rather a pre-existing trait"

Cross-comments from referee 2:

The comments of reviewer 1 are valid but I think the authors can address most of them with additional statistical analyzes as well as some reframing of their paper, and probably some additional experiments to boost the n value and hopefully reach statistical significance.

Therefore, I still recommend that the paper be reconsidered after major revisions requiring re-review.

Referee #1:

The work of Grieco and coworkers is focused on the causal relationship between social dominance, anxiety, stress resilience and neurogenesis. Although a number of papers have been published on the association of dominance with anxiety/stress vulnerability, as well as association between anxiety and neurogenesis, the current manuscript addressed the relationship between dominance and neurogenesis. Based on experiments in the first part of the manuscript, the authors claim that dominant individuals have lower adult neurogenesis than submissive individuals. The second part of the work tests whether reducing neurogenesis increases dominance status. Although the idea behind the possible link between dominance rank and neurogenesis and vice versa is novel and interesting, the data do not seem to support this connection. There are also technical issues with the neurogenesis manipulation experiments.

We thank this reviewer for his/her insightful review of our manuscript and for helping us to improve it.

Major points.

1. It is stated (line 106-108) that dominant individuals have lower BRDU+ cell counts in the dorsal DG (i.e., reduced neurogenesis), but corresponding Fig. 2G and corresponding legend do not show a neurogenesis difference between Control dominant and submissive mice (there is no posthoc test that would indicate a difference). Further, even the rank effect is trend only (indicated by a * that usually signals significance). In any event, the rank effect includes both Control and Stress mice, but the important groups to support the claim are the Controls because they are used to justify the neurogenesis phenocopy experiment.

We agree with the reviewer. To clarify the manuscript and to answer this reviewer's and the other two reviewers' comments, we have removed the complete dataset on chronic restraint stress, and we conducted a new experiment, with an increased sample size, to assess the levels of adult neurogenesis and anxiety before and after the establishment of social hierarchy. This new experiment confirmed our initial observation that future dominant mice display less neurogenesis in the hippocampus than the subordinates. The new data also indicates that neurogenesis and anxiety before group housing can predict hierarchical rank once mice are grouped.

The new data can be found in Figure 1 and Extended View Figures 2 and 3.

2. Fig 2H. Same issue with DCX labeling intensity, except that rank effect (for Control and Stress) is significant; but again, the important comparison would be between dominant and submissive Controls by a posthoc analysis, which is not indicated in the Fig./legend.

As mentioned above, we have redone the experiment of Figure 1 with an increased sample size and without stress resilience experiments.

In the new experiment, we have focused on cell proliferation, so we do not display data on DCX.

3. Since neurogenesis between dominant and submissive Controls does not seem to be different, justification to manipulate neurogenesis by TMZ is unfounded. Beyond this problem, TMZ causes a massive reduction, around 50% in neurogenesis, while the "non-significant" difference between Control dominant and submissive is around 15% (judging from the figure). Therefore, the TMZ experiment does not really phenocopy the dominance associated (but not significant) reduction in neurogenesis in Control mice.

While the TMZ experiment resulted in a significant reduction of around 50% in neurogenesis, the difference between dominant and submissive mice, is around 15% in the new dataset. However, the aim of the study was not to phenocopy the observed difference, but rather to investigate the role of neurogenesis in social dominance. The strong reduction of cell proliferation induced by TMZ injections therefore provides an important starting point to demonstrate the importance of adult neurogenesis in the regulation of hierarchy, that may be followed up with more subtle interventions.

4. The experimental design of the TMZ experiment is not optimal. In a proper design that tests causality between neurogenesis and social status, subordinates are selected for TMZ to see if chemically reducing

neurogenesis converts these animals dominant. Dominant mice should not respond to TMZ as they cannot be more dominant.

This is a very interesting experiment suggested by the reviewer and we agree that the experimental design of the TMZ experiment could be modified to test whether inhibiting neurogenesis in subordinates within a well-established hierarchy may increase their rank. However, in order to ask this question, one has to first demonstrate that adult neurogenesis is involved in the regulation of hierarchy in groups that have not established it, yet, which is the subject of the present study.

We have nonetheless mentioned this point in the discussion (paragraph 3, 3 last lines): ‘While our findings clearly establish the role of adult neurogenesis in the initial formation of social hierarchies, its involvement in the maintenance and regulation of dominance behavior within an already-established hierarchical group remains to be explored.’

5. Inhibition of neurogenesis by TMZ is unlikely to be limited to the dorsal DG, undercutting the association between dorsal DG neurogenesis and dominance. There are region specific manipulations of neurogenesis that are superior to chemical ablation.

We agree with the reviewer’s point. Furthermore, TMZ also inhibits neurogenesis in the olfactory bulb (OB), possibly interfering with social recognition, which may be important for hierarchy. In the new experiment presented in Figure 1 (and E.V. Fig. 2,3), we did not distinguish between dorsal and ventral neurogenesis. Furthermore, the TMZ experiment presented in Fig. 3 shows that in mice that had never met before, dominance in the SCTT is increased in animals injected with TMZ. Thus, social recognition (and therefore OB neurogenesis) does not seem to play a major role in the establishment of dominance in this design. However, we cannot formally exclude the participation of OB neurogenesis in the dominance behavior. We are now mentioning this point in the discussion (Paragraph 4, 3 last lines).

Additional comments

1. The introduction is a little rambling and difficult to follow. Anxiety is linked to dominance but then to subordination. Perhaps the authors wanted to say that the link is tenuous and both increased and reduced anxiety and stress susceptibility have been linked to dominance status. It would be useful to reorganize the intro and outline a clear rationale for the studies. Introduction of stress is particularly confusing as the paper is not about stress but rather about the relationship between social rank and neurogenesis. The purpose of Stress mice in this context is unclear. Also, this makes the first and second parts of the manuscript disjointed.

We agree with the reviewer's concern regarding the purpose of the stress in our study. To enhance the clarity of our manuscript and to answer point 1 of this reviewer, we have removed the complete dataset on chronic restraint stress. Additionally, we conducted an experiment to assess the levels of adult neurogenesis under baseline conditions in dominant versus subordinate mice. These results are shown in Figure 1 of the revised manuscript and indeed provide further rationale for the second part of the study involving the TMZ experiments. In view of this new data, we have modified and clarified the introduction.

2. Why are BRDU numbers are per slice in Fig. 1 while in the TMZ experiment in Fig. 2 the numbers are normalized to volume. The latter would be much better as mismatch of sections across animals and groups can introduce confounds.

We have now expressed the BrdU, IdU and CldU data in “cells/mm³”.

3. Full statistical analyses should be presented with posthoc data in the fig legends. This would help to understand whether changes are significant or not.

All statistical analyses are now presented in the figure legends.

4. It seems that BRDU positive cells in the ventral DG is higher in Control dominant vs. submissive mice (Fig. 2C), which is unusual as it would be the opposite to the claimed difference in the dorsal DG. Moreover, this may contribute to the CRS induced reduction in neurogenesis in dominant but not submissive mice.

We agree with the reviewer that this data was not clear and also did not lead to a clear explanation of the interaction between neurogenesis, stress resilience, anxiety and hierarchy. With the aim of clarification

and to answer this reviewer's point Nb. 1, we have performed another experiment with increase sample number and without CRS which, we believe, clarifies the main message of the manuscript.

5. Fig 2I. Correlation is shown for BRDU. What about DCX? If no correlation, why?

As mentioned above, the DCX data have been removed and we carried out correlations between social status under baseline conditions and IdU and CIdU positive cells in the hippocampus. Our new experiments show that baseline neurogenesis negatively correlated with social dominance behavior and pre-existing anxiety.

6. Calculation of anxiety score is not clear. It is mentioned that normalized values are used from 3 anxiety tests (time spent in dark/closed ...). Does this mean that dark/closed time was used or that light/open time was normalized to dark/closed? Was this approach of calculating anxiety validated in a well-established anxiety model? I found no reference.

The anxiety score was calculated using normalized values for the dark compartment, closed arms, and thigmotaxis. Importantly, similar results would have been obtained if we had used the time spent in the lit compartment, open arms, and center, as these measures inversely reflect the same anxiety-related behaviors, providing a consistent representation of the animals' anxiety levels.

To be more transparent regarding these data, all individual parameters used to generate the anxiety score are now provided in supplementary data (Fig. EV. 2) . Also, the anxiety scores are described in more detail in the methods section.

7. The authors claim that dominance is established gradually once animals are cohoused from 6 weeks of age. I assume that mice were cohoused with others from weaning to 6 weeks of age that could be sufficient to develop the dominant and submissive phenotypes. Dominant mice likely continue to be dominant in a new group. This is supported by published data showing that winners keep winning. Do the authors have evidence that experience prior cohousing at 6 weeks of age has no effect on dominance?

Our animals were purchased from Janvier and from weaning to their arrival in our facility (at 6 weeks of age), their individual life history is not available. Determining the precise age at which dominance behavior appears in mice is complex due to its multifactorial nature. The onset of dominance behavior is closely tied to sexual maturation which is around 35 days to 65 days of age (PMID 31541122). Another study has found that mice can form relatively stable hierarchies as early as weaning age (around 21 days). Life events are known to influence adult neurogenesis as well as many aspects of an individual's behavioral traits, possibly also dominance behavior. Thus, answering such a question would require following the life history of each individual, from birth to adulthood, and assessing events such as individualized maternal care, interactions between pups of the same litter, etc... While such a question is currently being explored in our lab, it largely extends beyond the scope of the present study. We discuss the question of the effect of life events on hippocampal neurogenesis in the emergence of individual traits (discussion, end of paragraph 7).

8. The number of animals in each group should be specified.

The number of animals per group for each experiment is now provided in the figure legends.

9. Label for the dominant and submissive groups should be in Fig2, like in Fig 1.

The labels for Dominants and subordinates are now homogenous throughout all figures.

Referee #2:

Summary:

Hierarchical dominance behavior in mice can be associated with higher trait anxiety and emerges in response to different stressors. Here the authors set out to explore how adult hippocampal neurogenesis may mediate the interplay between anxiety levels and dominance behavior in mice. The authors used SCTT to establish dominance and assessed neurogenesis levels and did functional manipulations of neurogenesis levels. Their results show that dominant mice display higher anxiety scores and vulnerability to chronic restraint stress, as well as impaired adult neurogenesis compared to subordinate mice after CRS. Adult neurogenesis depletion increased hierarchical dominance as well as situational dominance, irrespective of anxiety scores, indicating that neurogenesis can modulate dominance behavior independently of anxiety levels.

We thank this reviewer for these insightful comments on our manuscript and for helping us to improve it.

Major points:

1- Rationale for the putative role of adult hippocampal neurogenesis in the establishment of dominance behavior could be clearer/ more developed.

In the revised version of the manuscript, we have modified and clarified the introduction and developed the rationale of studying the role of adult neurogenesis in dominance behavior.

2- The authors use an anxiety score which considers the animals' behavior across different anxiety assays. The behavioral measures assessed in each of these assays should be included in supplemental information. Additionally, while this score can provide an account of the underlying anxiety of the animal, an explanation of the tests chosen and of the measures used for each test should be included. Namely, why use measures of time spent in the "safe" feature of the apparatus for each paradigm (closed arm in EPM, dark chamber in LDT, walls in the OFT/NO) as opposed to time engaging in exploratory behavior in the more anxiogenic areas of the respective tests. Finally, please clarify why are different combinations of tests used in each experiment to generate an anxiety score.

The values for each anxiety test are now shown on EV Figure 2, 4 and 5.

The anxiety score was calculated using normalized values for the dark compartment, closed arms, and thigmotaxis. Importantly, similar results would have been obtained if we had used the time spent in the lit compartment, open arms, and center, as these measures inversely reflect the same anxiety-related behaviors, providing a consistent representation of the animals' anxiety levels. To be more transparent regarding these data, all individual parameters used to generate the anxiety score are now provided in supplementary data.

In experiments shown in Figure 2 and 3, our goal was to comprehensively assess the effects of TMZ on anxiety-related behavior. The rationale for employing different combinations of tests was to increase the robustness of our findings by exploring whether TMZ would consistently influence anxiety-like behavior across a broader range of validated paradigms. While the open field, light-dark, and elevated plus maze tests are all well-established tools for assessing anxiety, we sought to include additional measures, such as the marble burying test in the second experiment, to account for potential variations in anxiety responses that different tasks might detect. This approach also allowed us to confirm the reliability of TMZ's effect by showing that any observed behavioral changes were not limited to a specific subset of tests but rather generalizable across multiple anxiety paradigms as showed by the individual behavioral measures in the supplementary data. Normalizing the results across three tests in each experiment ensures consistency in our scoring method, even as we explore diverse anxiety tests. Thus, the decision to vary the test combinations enhances the rigor of the study without compromising the validity of the anxiety score. This rationale is now explained in the material & methods section / Anxiety score / from line 9.

3- To claim that increased anxiety is a pre-existing trait in mice that become dominant, would require testing anxiety at baseline (e.g. single housing before cohousing, and testing anxiety levels before cohousing), and then observe if a priori innate anxiety predicts dominance behavior, and whether this is or not neurogenesis dependent. As is, the authors observed consistent higher anxiety levels once the animals had already been cohoused 1 or several weeks and established dominance, but not a priori trait

anxiety.

To answer the comment from this reviewer and from the other 2 reviewers, we performed a new experiment aimed at assessing the levels of anxiety and of adult neurogenesis before and after social hierarchy establishment in dominant versus subordinate mice. In this new dataset, we confirmed and strengthened our initial observation with an increased sample size, that future dominant mice display less neurogenesis in the hippocampus than the subordinates along with heightened anxiety levels, prior the establishment of social hierarchy. This new set of data, now shown on Figure 1 of the revised manuscript, provides further rationale for the second part of the study involving the TMZ experiments.

4- Regarding the hypothesis that a priori levels of neurogenesis influence dominance behavior: this is not clearly addressed with this experimental design, as the timing of BRDU injections is always after cohousing and manipulations have started and may be influencing the levels of new neurons labelled and their survival and maturation. Without live imaging throughout the timeline, it is hard to make claims about a priori levels of neurogenesis. The discussion should clarify this.

Indeed, live imaging is an outstanding way to follow neurogenesis over a long period of time. However, this technology is cumbersome and certainly not adapted to the aggressiveness that occurs between individuals during the establishment of hierarchy.

To improve the experimental design and try to address the point raised by the reviewer, as presented in our answer to the question above, we used injections of two thymidine analogs before (IdU) and after the establishment of hierarchy (CldU). These results are presented in the new Figure 1 and show that cell proliferation before group housing predicts hierarchy with a precision of 81%. Although the number of IdU- and CldU- labeled cells can be influenced by events occurring during the 9 and 3 weeks after the respective analogs injections, both labeling indicate that incorporation is lower in dominant animals throughout the experiment. Thus these results indicate that hierarchy itself does not influence neurogenesis but rather, neurogenesis influences hierarchy, as confirmed by the TMZ experiments shown in Figures 2 and 3.

5- To manipulate neurogenesis, the authors used TMZ, which affects olfactory bulb neurogenesis as well as hippocampal neurogenesis, as if referred in the discussion. It would be crucial, for the experiments that utilize this treatment, to assess the effect of TMZ in odor discrimination and in social recognition.

We agree that TMZ likely also inhibits neurogenesis in the olfactory bulb (OB), that is involved in social recognition. As commented by Reviewer 1, assessing the role of hippocampal versus OB neurogenesis in dominance would require other ablation methods, such as X-ray irradiation, which is out of the scope of the present study. However, the experiments presented in Figure 3 show that TMZ increases dominance on mice that had never met before, suggesting that social recognition does not play a major role in this experimental design.

This point is raised and commented in the discussion (end of paragraph 4).

6- Since the hippocampus is involved in social memory and social recognition, namely through its CA2 subregion, it would be important to also control the effects of manipulations of hippocampal neurogenesis in a task of social memory, to see if disrupted social memory (recognition and discrimination) could explain the effect of neurogenesis in dominance behavior.

We agree with the reviewer that social memory likely plays a role in the SCTT used here to assess hierarchy. However as commented in the point 5 above, Figure 3 shows an effect of TMZ in absence of social memory, since it tested animals that had never met before the SCTT. Although this experiment does not exclude an effect of social memory on hierarchy, it shows that dominance can occur in its absence. Assessing the role of social recognition on dominance would require experimental designs that are out of the scope of the present study.

As in point 5, the role of social memory on dominance is commented in the discussion (paragraph 4, 7 last lines).

Minor points:

1-Introduction: a review article (reference 1) is used multiple times instead of the specific relevant research articles.

This has been modified

2-Figure 2:

- For DCX, since the measure used was intensity instead of cell counts, elaborate on how intensity was measured and normalized.

To enhance the clarity of our manuscript, we have removed the complete data set on chronic restraint stress (CRS), that included DCX counts. Instead, we conducted an experiment to assess the levels of adult neurogenesis under baseline conditions in dominant versus subordinate mice. This was done to provide further rationale for the second part of the study involving the TMZ experiments.

3- Here the correlation analysis was only shown for mice post CRS. The authors could also show analysis in the control animals, even if only in supplement; likewise for analysis of DCX levels vs. anxiety scores. As mentioned above, to enhance the clarity of our manuscript, we have removed the complete data set on chronic restraint stress (CRS), as these experiments were not sufficiently powered. Instead, we conducted a new experiment to assess the levels of adult neurogenesis under baseline conditions in dominant versus subordinate mice, with an increased sample size. This was done to provide further rationale for the second part of the study involving the TMZ experiments. In the revised manuscript, correlation analyses have been conducted in both group dominant and subordinate mice, as can be seen in Fig. 1I,J and EV Fig. 3C

4- To further corroborate their point, using the data already collected, the authors could analyze the correlation between anxiety scores and rank level, and between neurogenesis levels and rank level; or compare the animals that rank 1st vs animals that rank 4th.

This is now done on the new dataset of Fig.1 and EV Fig. 3.

5-Figure 4:

- The experimental design is not fully clear. Please clarify if the data displayed are for the first and second or only the second round of tests, after determining "innate" dominance of mice in both control and TMZ groups and then making new pairs? How were the animals matched for innate anxiety if testing of anxiety was done after the SCTT - was there a previous batch of anxiety tests done before the second round of testing? Please elaborate on the timeline of this experiment.

One week after arrival, we first assessed weight and anxiety in mice using an elevated plus maze, a light dark test and an open field test. We then used this information to form the groups to obtain anxiety- and weight-matched groups (see results below, data not shown in the manuscript). After this, mice were treated with either TMZ or NaCl and then assessed on the SCTT (data shown on Fig. 3B-D), followed by another set of anxiety of tests (data shown on Fig. 3E).

This is now clarified in the results section, paragraph 6, lines 6-8.

6-1Typos:

Line 61: "an observational study [and] two interventional studies"

Line 129: "SCTT test" -> "SCTT"

Line 212: ablation or reduction?

Line 282: "in human[s]"

Line 488: "by using the number of [DAPI or NeuN] positive cells"?

Thank you. This is now corrected.

In conclusion:

This manuscript presents very interesting new data which shows that dominant mice display higher levels of anxiety, and that neurogenesis levels differ between dominant and submissive mice (in the dDG) and reflect their different susceptibilities to stress (in the vDG). Additionally, manipulations of neurogenesis in a non-targeted way affected dominance behavior independently of anxiety levels, but this effect remains to be explained. To support its main claims further justification and additional experiments are required. I advise major revisions requiring re-review.

Referee #3:

The authors examined the role of adult neurogenesis in linking social rank and anxiety level. This is a very interesting study, but the way it is presented does not render justice to the very nice data of this work. Indeed adult neurogenesis has been shown to "regulate" social hierarchy in one hand and anxiety-like behaviour in another one. In the present paper, both traits were examined in the same animals in basal condition and in response to chronic stress. This is a novel aspect of the story that is underexploited in the manuscript. Just to give an example: the title: "Adult neurogenesis regulates social dominance and anxiety". The fact that adult neurogenesis regulates these traits is already known.

The way the manuscript is written is complex and often does not correspond exactly to the experiments carried out. For example (L60), the authors write that they will investigate "whether inter-individual differences in adult neurogenesis may underlie differences in anxiety and dominance behavior". The design of the experiment (between-group analysis rather than a within-group analysis) does not allow to study inter-individual differences per se.

It's also hard to know what message to retain. For example (L22) the authors wrote "these results indicate that adult neurogenesis regulates hierarchical and situational dominance behavior along with anxiety-related behavior, independently from each other. This conclusion is unclear as in figure 3 it is shown that reduction of cell proliferation is impacting both traits. In Figure 4, animals are extremely anxious which explain the lack of effects on anxiety.

We thank this reviewer for these insightful comments on our manuscript and for helping us to improve it. We have changed the title of the manuscript, to better reflect the data, including the new data of Figure 1 (see point 18 below). We have also clarified the introduction and the discussion.

METHODS

1•The experimental design should be described after the description of the tasks!

This has been modified in the methods section.

•In this study, adolescent animals (7 week at the beginning of testing) were used. It is understandable that working with adult male mice (10-12 weeks) is complicated by their aggressive behaviour. However, it would be appropriate to state in the text why such young animals were chosen.

Indeed, we chose young males to avoid excessive display of aggressivity. This is now stated in the materials & methods section / Animals / line 2.

2•L318 Add the number of animals per experiment and per group. Are the number of animals similar in Fig1C left and right panel and in Fig1 D,E

To enhance the clarity of the manuscript and to answer comments from the two other reviewers, we have removed the complete dataset on chronic restraint stress (CRS), i.e. Fig. 1 and 2 and replaced it by a more carefully designed experiment aimed at testing whether adult neurogenesis and anxiety before group formation are related to and can predict the hierarchical rank. In this new experiment (now Fig. 1 and EV Fig. 2, 3), we have also included more mice, in order to gain statistical power. The number of animals for each experiment is presented in the corresponding legends.

3•L467 The authors wrote "chosen at regular intervals ". Please indicate the sampling probability (1 in X). We have assessed 1 out of 6 sections. This is now stated in the methods section (methods section, Immunofluorescence staining, line 1)

4•L485 why the number of Dcx cells were not counted manually? What does mean DCX labelling intensity ? This approach can lead to misleading results.

As mentioned above, this experiment has been replaced by a more carefully designed experiment, in which we did not analyze DCW, but instead, focused on the incorporation of thymidine analogs at different time points (please see Fig.1).

5•L487: It is indicated that the volume density has been measured. However, the results are expressed as nb of cell/section. The quantification of cells (nb of cell/section) is unconventional in the field

(considered as biased quantification). Therefore, I would recommend to present the results in volume density or to use the optical fractionator, as the number of cells was evaluated along the septotemporal axis (Nb of cells = sum of nb of cells/section X the inverse of the sampling probability).

All the immunohistochemistry data is now presented as the amount of cell/volume.

6•L322: EPM, the OF/NO test, and the LDT were measured in experiment 1. The results for each task should be detailed in addition to the anxiety score. Were mice hyperactive? Note that on the figure 1, EPM has been forgotten!

To be more transparent regarding these data, all individual parameters used to generate the anxiety score are now provided in supplementary data (EV Fig. 2, 4, 5). Note that we have removed the complete data set on chronic restraint stress (CRS), as mentioned above. We have also measured locomotor activity in the TMZ-treated mice and found no difference between group (see data below).

7•L324 How the mice were separated into two groups? Were the anxiety score of the control and the stress group similar before CRS?

As mentioned above, we have removed the complete data set on chronic restraint stress (CRS). However, in the new dataset, the anxiety score was assessed before group formation to avoid any bias. The results of this test is shown in the new Figure 1, panel B.

8• L329 The marble burying test (MBT) is measuring anxiety

Thank you. This has been modified in the text (methods section, experimental design, experiment 2, line 8)

RESULTS

9•L78: results of the plus maze are lacking

As mentioned above, we have replaced Fig. 1 with a new experiment. In this new experiment, we provide all data on all behavioral tests in EV Fig. 2.

10•L99: adult neurogenesis has been shown to be involved in anxiety-like behaviour (Revest et al.2009). Note that the authors confirmed this conclusion using TMZ.

Thank you. Indeed, this study inspired us to assess the role of adult neurogenesis in dominance behavior. This point is clarified in the introduction. We are mentioning this publication in the results section (Paragraph 3, line 1) and the discussion (Paragraph 7, line 15).

11•L94: The authors wrote: "we observed that after CRS, dominant mice displayed decreased stress resilience compared to subordinate mice. This was reflected by an increased number of buried marbles in the marble burying test (Fig. 1D) and the development of social avoidance toward an unfamiliar CD1 mouse in a social interaction test (SI) (Fig. 1E). In contrast, subordinate mice responded similarly to unstressed mice in both tests, suggesting a resilient phenotype against chronic restraint stress (Fig. 1D, E). It is a convoluted way of saying that CRS causes further anxiety in the dominant but not in the subordinate.

Indeed. However, this experiment has been replaced.

12•L118 Inhibiting adult neurogenesis increases social dominance behavior and anxiety

We have modified the title accordingly.

13• Are anxiety score before and after CR correlated (within each group)

To enhance the clarity of our manuscript, we have removed the complete dataset on chronic restraint stress (CRS).

14• Are anxiety score before CR predictive of the level of adult neurogenesis (within each group) To enhance the clarity of our manuscript, we have removed the complete data set on chronic restraint stress (CRS). However, we performed this analysis in the new experiment presented in Fig. 1, and found that anxiety and adult neurogenesis were both predictive of hierarchy.

15• The effects of CRS on adult neurogenesis are less obvious. Isn't this due to the housing conditions: stressed animals are housed with control animals (a procedure not typically used in behavioural studies) which could buffer the effects of stress. This needs to be discussed.

This data has been removed in the revised version of the manuscript.

16• Figure 3 and 4: please provide individual performances in the EPM, the OF/NO test, and the LDT

All individual values used to generate the anxiety score are now provided in supplementary data. Please see Figures EV 2, 4, 5.

17• Figure 4: to which extent adult neurogenesis (Dcx) was reduced?

We have not done histological assessment of neurogenesis in this figure. However, since the conditions, the timeline and the TMZ concentrations were identical, we expect that the effect of the TMZ treatment is the same in Figure 3 (former Figure 4) as in Figure 2 (former Figure 3).

DISCUSSION

18• LN 173. The authors wrote we examined the role of adult hippocampal neurogenesis in shaping social dominance behavior.

This would have been the case if neurogenesis was measured before measuring social dominance behavior. Rather their findings revealed that the level of anxiety predicts social rank that influence vulnerability to CRS: subordinates are more resilient to stress compare to dominants.

In the new experiment presented in Figure 1, we assessed the level of adult neurogenesis and anxiety under baseline conditions (i.e. before) and after social hierarchy establishment in dominant versus subordinate mice. This was done to provide further rationale for the second part of the study involving the TMZ experiments. We confirmed our initial observation with an increased sample size mainly that future dominant mice display less neurogenesis and increased anxiety in the hippocampus than the subordinates. Furthermore, we found that both anxiety and neurogenesis predict the hierarchical status.

19• Line 187 "but is rather a pre-existing trait"

This is now corrected

Cross-comments from referee 2:

The comments of reviewer 1 are valid but I think the authors can address most of them with additional statistical analyzes as well as some reframing of their paper, and probably some additional experiments to boost the n value and hopefully reach statistical significance.

Therefore, I still recommend that the paper be reconsidered after major revisions requiring re-review.

Thank you for this comment. We present new data in Figure 1, together with new statistical analyses (i.e. predictive value of neurogenesis and anxiety on hierarchical status), and increased sample size.

Dear Prof. Toni,

Thank you for the submission of your revised manuscript. We have now received the enclosed reports from the referees as well as cross-comments from referee 3, all pasted below.

As you will see, referee 1 still has a few concerns that I would like you to address before we can proceed with the official acceptance of your manuscript. I asked referee 3 for cross-comments on referee 1's report and I think referee 3's suggestions are good. Especially the concern on pre-existing social hierarchy needs to be addressed. Please let me know if you have any questions regarding the revisions and we can discuss this further, also in a video chat, if you like. Please also address all other referee concerns and co-submit a final point-by-point response with your final ms.

My feeling is that we can use "predict" in the ms title, instead of "correlate", but I will also ask my colleagues for feedback. Please do write the abstract in present tense, as per journal policy.

A few editorial requests will also need to be addressed:

- Your ms has 3 main figures and should thus be published as a short report with combined results and discussion sections.
 - Please add up to 5 keywords to the ms file.
 - Please add a "Disclosure and Competing Interest Statement" to the ms file.
 - Please list the corr. author's email address on the ms title page.
 - The author credits need to be removed from the ms file. All credits are entered during online ms submission.
 - Please correct the reference format to EMBO reports (Harvard) style: it needs to be alphabetical, not numerical; et al needs to be used after 10 author names; DOIs should only be used for preprints and datasets that have not been published yet
 - Please remove "data not shown" on page 5 as per journal policy.
 - Please add the missing callout for Fig. 1F.
 - A Reagents and Tools Table (listing key reagents, experimental models, software and relevant equipment and including their sources and relevant identifiers) needs to be added to the method section as individual file. A downloadable templates (.docx) for the Reagents and Tools Table can be found in our author guidelines: <<https://www.embopress.org/page/journal/14693178/authorguide#manuscriptpreparation>>.
 - The source data (SD) for Fig 2A is missing, please add. The SD for the main figures need to be uplidd as 1 folder per figure, while the SD folders for the EV figures need to be grouped into one zipped folder. Please also enter the SD information in the SD checklist.
 - The manuscript sections should be in the following order: Title page - Abstract & Keywords - Introduction - Results - Discussion - Methods - Data Availability - Acknowledgments - Disclosure Statement & Competing Interests - References - Figure Legends - (Main Tables with legends) - Expanded View Figure Legends.
 - Materials and methods should just be Methods
- Please address the following comments on the *Figure Legends*:
- Please note that the legend for figure EV 2 (B-E-H, C-F-I, D-G-J) are not provided in the sequential manner. This needs to be rectified.
 - Please note that sub-figure F is missing from figure EV 3 in the manuscript. This needs to be rectified.
 - Please note that the exact p values are not provided in the legends of figures 1G, 2B, 2F, G; 3D, EV1 C, D.
 - Please indicate the statistical test used for data analysis in the legends of figures 2C, 3E.
 - Please note that * is not defined in the legend of EV3 A, C. Kindly rectify the same.
 - Please note that # is not defined in the legend of figure 3D. Kindly rectify the same.
 - Please note that information related to n is missing in the legends of figures 3E, EV1 H.
 - Although 'n' is provided, please describe the nature of entity for 'n' in the legends of figures EV1 C, D, E; EV2 B-J; EV3 B-D; EV4 B-D.
 - Please note that the error bars are not defined in the legend of figure EV2 B-J.
 - Please note that the white arrow heads are not defined in the legend of figure 1F. This needs to be rectified.

- Please note that axis gaps are not labeled appropriately in figures EV2 B-J, EV3 B-D, EV4B-D.

EMBO press papers are accompanied online by A) a short (1-2 sentences) summary of the findings and their significance, B) 2-3 bullet points highlighting key results and C) a synopsis image that is exactly 550 pixels wide and 200-600 pixels high (the height is variable). The synopsis image should provide a sketch of the major findings, like a graphical abstract. Please note that text needs to be readable at the final size. Please send us this information along with the final manuscript.

Referee #1:

Although the authors rewrote the paper and dropped the confusing stress related data, some of the previous issues have not been fully resolved.

The major concern is the so called "preexisting" anxiety and neurogenesis at 6 weeks of age predicting future dominance behavior. The problem with this interpretation is that it ignores the fact that juvenile pups, at P28, already formed dominance hierarchies that is stable in the same group up to P42 (6 weeks of age) and beyond to adulthood (<https://doi.org/10.3389/fncir.2021.676308>). Therefore, "predicting future dominance" is misleading, as the 6-week anxiety and neurogenesis measures reflect the current dominance rank and not that will develop later (as the authors suggest). It is unlikely that one week of isolation alters the individual's dominance rank and dominant individuals will occupy high rank even in a new group. Therefore, 6-week measures reflect a correlation between neurogenesis, anxiety, and dominance rank, but unfortunately dominance was not measured at this early age. This issue requires to change the title to avoid confusion.

Further, while the authors claim a neurogenesis difference at both 6 and 12 weeks of age, the representative images show no neurogenesis difference btw dominant and subordinate at 12 weeks of age. Further, although the images show far less CldU then IdU cells, which is consistent with reduced neurogenesis at 12 weeks as compared to 6 weeks of age, the summary graphs curiously show similar numbers per mm² for the two labels that is not possible for the above reason.

Inhibiting neurogenesis by TMZ is nonspecific and its robustness (50%) may cause a lot more changes than the natural difference in neurogenesis (10%). For example, given that young neurons, via interneurons, provide feedback inhibition to the dentate gyrus, a slight, physiologically relevant reduction in this feedback is totally different than a massive 50% reduction. Not even stress results in such drastic change that could lead to network adaptation and consecutive circuit and behavioral changes that are not seen with under physiological circumstances. Since 12 TMZ injections were used, the dose could have been titrated to mimic naturally occurring differences in neurogenesis.

Explanation for using the closed arm as a measure of anxiety is still not satisfying as this unusual, and it makes the data difficult to compare to almost any existing data. Moreover, recordings and calcium imaging from the hippocampus (relevant because of neurogenesis) show that closed and open activity are not the opposite of each other indicating that they measure different aspects of anxiety. But in any case, if closed shows the same change than open activity (according to the authors), why not use the conventional approach? Or at least show how closed and open measures are correlated.

Other comments

The abstract and introduction needs further improvement, mostly linguistic, beside the interpretational issue Fig. 1. Legend lacks details and it is difficult to correspond the legend with the graphs. With some effort, one can figure it out, but it should be clearly described in the legend. Also, how was the population mean calculated (I assume that it is the dashed line at 0.5, but it is not explained in the legend).

Line 140

It is incorrect to state that single housing prevents the formation of social hierarchy because it has already been formed by that time (see above)

Line 173

"even in the absence of prior social experience" This statement is incorrect as the mice had social experience up to 6 weeks of age. In fact, there should be a section in the discussion how environmental factors such as maternal environment, play etc.

during the juvenile period modulate behavior and produce individual differences.

Line 209

it is stated that the neurogenesis effect on dominance is mediated by anxiety. This is confusing because in several experiments anxiety is unchanged

In light of the interpretation problem with the notion of preexisting anxiety and neurogenesis, discussion should be rewritten

Overall, discussion is confusing and is way too long. It would be useful to have a model describing the findings in an easily understandable format

The result section should be segmented to paragraphs to better separate individual experiments and thoughts

It would also be useful to use other forms of dominance measures

Referee #2:

The authors have addressed my comments satisfactorily

Referee #3:

Grieco et al have extensively revised their manuscript and the present version is much clearer and very convincing.

The only point that need to be corrected is that adolescent AND adult neurogenesis have been studied. This is an important point for the field (see Arenallo&Racic,2024). In fact, the mice were injected with CldU at 6 or 7 weeks, late adolescence. I think this adds another dimension to the results! Adolescent neurogenesis predicts adult neurogenesis (a correlation between Idu and Cldu cell numbers would be welcome) and the "developmental trajectory" of the animals. A few lines in the discussion on this point (adol versus adult) would be appreciated.

The text need to be corrected accordingly. For example :

Title "Natural variations of adolescent neurogenesis and anxiety predict hierarchical status of inbred mice Line 194: "adult" should be replaced by " adolescent" etc..

Minor

line 72 it is unclear if anxiety was tested in 6-week-old mice or in 7-week-old mice (see methods line 288)

What was the delay between the 3 anxiety tests?

It is unclear how old were the animals when injected with Idu? 6 or 7 weeks?

Typos in the text and the graphs (EV2 and : closAed arms)

Cross-comments from referee 3:

> Referee #1

>

> Although the authors rewrote the paper and dropped the confusing stress related data, some of the previous issues have not been fully resolved.

>

> The major concern is the so called "preexisting" anxiety and neurogenesis at 6 weeks of age predicting future dominance behavior.
> The problem with this interpretation is that it ignores the fact that juvenile pups, at P28, already formed dominance hierarchies that is stable in the same group up to P42 (6 weeks of age) and beyond to adulthood (<https://doi.org/10.3389/fncir.2021.676308>). Therefore,
> "predicting future dominance" is misleading, as the 6-week anxiety and neurogenesis measures reflect the current dominance rank and not that will develop later (as the authors suggest). It is unlikely that one week of isolation alters the individual's dominance rank and dominant individuals will occupy high rank even in a new group. Therefore,
> 6-week measures reflect a correlation between neurogenesis, anxiety,

- > and dominance rank, but unfortunately dominance was not measured at
- > this early age. This issue requires to change the title to avoid
- > confusion.

I do not fully agree with the reviewer. The authors assess neurogenesis at two developmental stages, adolescence and adulthood (see my previous review), and even if the hierarchy is already established in adolescence, the fact that levels of neurogenesis at this time correlate with dominance rank in adulthood is very interesting. However, the authors should modify their text by removing the "notion of preexisting anxiety and neurogenesis" and introducing the adolescent period

I would change the title as following:

Natural variations of adolescent neurogenesis and anxiety is associated (-or is correlated) to the hierarchical status of adult inbred mice : to please the reviewer

personally I would propose: Natural variations of adolescent neurogenesis and anxiety predict the hierarchical status of adult inbred mice

- > Further, while the authors claim a neurogenesis difference at both 6
- > and 12 weeks of age, the representative images show no neurogenesis
- > difference btw dominant and subordinate at 12 weeks of age. Further,
- > although the images show far less CldU then IdU cells, which is
- > consistent with reduced neurogenesis at 12 weeks as compared to 6
- > weeks of age, the summary graphs curiously show similar numbers per
- > mm² for the two labels that is not possible for the above reason.

A better illustration could be chosen.

I think that correlating the number of Cldu at the 2 time points (same for ldu) will answer the reviewer's question. I notice that the authors did not explain that they separated the 2 subgroups at the median to run a group analysis. This is not the best way: they should select the extreme quartile of the population.

- > Inhibiting neurogenesis by TMZ is nonspecific and its robustness (50%)
- > may cause a lot more changes than the natural difference in
- > neurogenesis (10%). For example, given that young neurons, via
- > interneurons, provide feedback inhibition to the dentate gyrus, a
- > slight, physiologically relevant reduction in this feedback is totally
- > different than a massive 50% reduction. Not even stress results in
- > such drastic change that could lead to network adaptation and
- > consecutive circuit and behavioral changes that are not seen with
- > under physiological circumstances. Since 12 TMZ injections were used,
- > the dose could have been titrated to mimic naturally occurring
- > differences in neurogenesis.

It is known that TMZ like other cancer drugs (arac ..) are non-specific and mimicking a naturally occurring differences in neurogenesis using TMZ is an illusion.

Sacrificing another cohort of mice to answer this question, which will not change the take home message, seems pointless to me

Moreover , if the extreme quartiles are considered, differences in cell numbers between subordinate and dominant are in between 20% to 40%.

- > Explanation for using the closed arm as a measure of anxiety is still
- > not satisfying as this unusual, and it makes the data difficult to
- > compare to almost any existing data. Moreover, recordings and calcium
- > imaging from the hippocampus (relevant because of neurogenesis) show
- > that closed and open activity are not the opposite of each other
- > indicating that they measure different aspects of anxiety. But in any
- > case, if closed shows the same change than open activity (according to
- > the authors), why not use the conventional approach? Or at least show
- > how closed and open measures are correlated.

I think that here s/he is asking to express the results differently : % time and % entry in the open arms compared to time in open arms + closed arms.

Referee #1:

Although the authors rewrote the paper and dropped the confusing stress related data, some of the previous issues have not been fully resolved.

1. The major concern is the so called "preexisting" anxiety and neurogenesis at 6 weeks of age predicting future dominance behavior. The problem with this interpretation is that it ignores the fact that juvenile pups, at P28, already formed dominance hierarchies that is stable in the same group up to P42 (6 weeks of age) and beyond to adulthood (<https://doi.org/10.3389/fncir.2021.676308>). Therefore, "predicting future dominance" is misleading, as the 6-week anxiety and neurogenesis measures reflect the current dominance rank and not that will develop later (as the authors suggest). It is unlikely that one week of isolation alters the individual's dominance rank and dominant individuals will occupy high rank even in a new group. Therefore, 6-week measures reflect a correlation between neurogenesis, anxiety, and dominance rank, but unfortunately dominance was not measured at this early age. This issue requires to change the title to avoid confusion.

As mentioned in the study referenced by this reviewer, a hierarchy that has formed before weaning can be destabilized during a critical period of adolescence in mice. In the present study, we assessed neurogenesis at two developmental stages: adolescence and adulthood and found that adolescent neurogenesis predicted the hierarchical status in adulthood. In the revised version of the manuscript, we have introduced the notion of adolescent neurogenesis (Lines 79-84) and modified the title to remove the notion of preexisting anxiety and neurogenesis. We have also modified the text at the end of the results section to mention that the role of adult neurogenesis in regulating pre-established hierarchies later in life still requires investigation (Lines 215-218). The modifications are highlighted in red in the revised version of the manuscript.

2. Further, while the authors claim a neurogenesis difference at both 6 and 12 weeks of age, the representative images show no neurogenesis difference btw dominant and subordinate at 12 weeks of age. Further, although the images show far less CldU then IdU cells, which is consistent with reduced neurogenesis at 12 weeks as compared to 6 weeks of age, the summary graphs curiously show similar numbers per mm² for the two labels that is not possible for the above reason.

We are now showing an illustration that better represents the difference in neurogenesis between dominants and subordinate animals (Fig. 1J).

For the quantification of CldU and IdU immunostained cells, one has to take into consideration that although both thymidine analogs label dividing cells, the number of labeled cells vary according to the analog and the antibody used (See for example Leuner et al. J.Comp.Neurol 2009 – PMID 19731267). Thus, a difference between the numbers of CldU and IdU-labeled cells

(or the lack thereof) does not necessarily reflect differences in the fate of the dividing cells. Here, we injected these two analogs to enable a comparison between groups, not between time points. Nonetheless, in order to compare the number of CldU and IdU cells and to assess a possible relationship between adolescent and adult neurogenesis, we used a correlation analysis between these 2 markers. This analysis indicates that adult neurogenesis correlates with adolescent neurogenesis, suggesting that natural variations in adult neurogenesis are long-lasting. This new analysis is now shown on Fig. 1M.

3. Inhibiting neurogenesis by TMZ is nonspecific and its robustness (50%) may cause a lot more changes than the natural difference in neurogenesis (10%). For example, given that young neurons, via interneurons, provide feedback inhibition to the dentate gyrus, a slight, physiologically relevant reduction in this feedback is totally different than a massive 50% reduction. Not even stress results in such drastic change that could lead to network adaptation and consecutive circuit and behavioral changes that are not seen with under physiological circumstances. Since 12 TMZ injections were used, the dose could have been titrated to mimic naturally occurring differences in neurogenesis.

We agree with this reviewer that TMZ is not specific and induces a stronger decrease in neurogenesis than the the average of the natural variations observed between dominants and subordinates. Although titrating TMZ may better phenocopy the natural variations in cell proliferation, such approach will not solve the specificity issue of the treatment and is out of the scope of the present study.

We note however, that the 50% reduction in neurogenesis induced by TMZ is comparable to the differences observed between the extremes of the natural variations. Indeed, when we compare the 4 mice with the lowest CldU counts and the 4 mice with the highest CldU counts of Fig. 1J, we obtain a 50% reduction in CldU. Interestingly, between these extremes, we observe a 50% reduction in anxiety score and a 65% reduction in the number of wins, which are very similar to the modifications induced by TMZ (Fig. 2C, E and G). Thus, both natural variations and TMZ data support the idea that adult hippocampal neurogenesis regulates anxiety and situational dominance behavior to a comparable extent.

4. Explanation for using the closed arm as a measure of anxiety is still not satisfying as this unusual, and it makes the data difficult to compare to almost any existing data. Moreover, recordings and calcium imaging from the hippocampus (relevant because of neurogenesis) show that closed and open activity are not the opposite of each other indicating that they measure different aspects of anxiety. But in any case, if closed shows the same change than open activity (according to the authors), why not use the conventional approach? Or at least show how closed and open measures are correlated.

There is no rule for choosing the open arm (OA) value over the closed arm (CA) value in assessing anxiety. Indeed, the CA value has been used in several studies, either alone (e.g: PMID 36610590), or in comparison with the OA value (e.g: PMID 36432096, 36422097). A correlation analysis between the OA and CA values of our experiments (see below) shows, as expected, a strong inverted correlation.

However, the data is not distributed along a perfectly inverted line. In fact, when looking at the proportion of total time spent in either arm, the CA time shows more variance than the OA time and therefore is more amenable to discriminate individuals than the OA time. We are now showing all the data we collected in the elevated-plus maze and the light-dark test in EVFigure 2.

Other comments

5. The abstract and introduction needs further improvement, mostly linguistic, beside the interpretational issue

The abstract and introductions have been clarified.

6. Fig. 1. Legend lacks details and it is difficult to correspond the legend with the graphs. With some effort, one can figure it out, but it should be clearly described in the legend. Also, how was the population mean calculated (I assume that it is the dashed line at 0.5, but it is not explained in the legend).

The legend in Fig. 1 has been clarified. Indeed, the dashed line at 0.5 represents the population mean, which is now explained in the legend, lines 661-662 of the revised manuscript.

7. Line 140

It is incorrect to state that single housing prevents the formation of social hierarchy because it has already been formed by that time (see above).

We have now rephrased this sentence to: "To assess the role of adolescent neurogenesis in absence of social memory, we tested the effect of TMZ in a second cohort of male mice which were singly housed." This new sentence can be found on lines 194-195 of the revised version of the manuscript.

8. Line 173

"even in the absence of prior social experience" This statement is incorrect as the mice had social experience up to 6 weeks of age. In fact, there should be a section in the discussion how environmental factors such as maternal environment, play etc. during the juvenile period modulate behavior and produce individual differences.

This sentence has been removed. We have also added a paragraph at the end of the manuscript, discussing the role of experience in shaping individuality. See lines 247-251 of the revised manuscript.

9. Line 209

it is stated that the neurogenesis effect on dominance is mediated by anxiety. This is confusing because in several experiments anxiety is unchanged

We have modified this statement as : "However, we observed that the correlation between anxiety and the proportion of wins in the SCCTT decreased over time and disappeared after hierarchy

*stabilization (after 5 weeks of cohabitation; Fig. 1E, Fig. EV3A, B), suggesting that social dominance may be independent from anxiety.”
This modification appears in lines 132-135 of the revised manuscript.*

10. In light of the interpretation problem with the notion of preexisting anxiety and neurogenesis, discussion should be rewritten

As mentioned above, the notion of preexisting neurogenesis and anxiety has been removed to introduce the notion of adolescent neurogenesis. This modifications appear throughout the manuscript, including the title. As per the journal's guidelines, the discussion section has now been merged with the results section.

11. Overall, discussion is confusing and is way too long. It would be useful to have a model describing the findings in an easily understandable format

The discussion has been modified and shortened. We have also added a schematic summary of our findings under the form of a graphical abstract.

12. The result section should be segmented to paragraphs to better separate individual experiments and thoughts

This has been done

13. It would also be useful to use other forms of dominance measures

Previous studies have shown that results in the tube test are relevant to other measures of dominance in the cage and can therefore replace them (see for example Fan et al 2019, PMID 30770887). We mention this in lines 114-116 of the revised manuscript.

Referee #2:

The authors have addressed my comments satisfactorily

Referee #3:

Grieco et al have extensively revised their manuscript and the present version is much clearer and very convincing.

1. The only point that need to be corrected is that adolescent AND adult neurogenesis have been studied. This is an important point for the field (see Arenallo&Racic,2024). In fact, the mice were injected with CldU at 6 or 7 weeks, late adolescence. I think this adds another dimension to the results! Adolescent neurogenesis predicts adult neurogenesis (a correlation between ldu and Cldu cell numbers would be welcome) and the "developmental trajectory" of the animals. A few lines in the discussion on this point (adol versus adult) would be appreciated.

The text need to be corrected accordingly. For example :

Title "Natural variations of adolescent neurogenesis and anxiety predict hierarchical status of inbred mice Line 194: "adult" should be replaced by " adolescent" etc..

We agree with this reviewer that we have examined the role of adolescent neurogenesis in determining dominance in adulthood. We have made the corresponding modifications in the title

and throughout the abstract and the manuscript. In particular, we have introduced the notion of adolescence as a critical stage for hierarchy plasticity (Lines 79-84 of the revised manuscript), modified the subtitles of the results sections, and the end of the results section (Lines 214-218) to reflect that the role of adult neurogenesis in regulating dominance behavior still requires further investigation. At the end of the manuscript, we have highlighted the role of early life experiences in shaping individuality as a possible mechanism underlying maturation variations in behavior and adult neurogenesis (Lines 247-251).

Minor

2. line 72 it is unclear if anxiety was tested in 6-week-old mice or in 7-week-old mice (see methods line 288)

Anxiety was assessed at 6 weeks, 8 weeks and 11 weeks of age. This is now clearly stated in the figure (See Fig. 1A) and in the methods section (Lines 278-287).

3. What was the delay between the 3 anxiety tests?

An interval of 24h was used between each test. This is now clarified in the methods section (Line 279-281).

4. It is unclear how old were the animals when injected with IdU? 6 or 7 weeks?

Animals were injected with IdU at PND 48. This is now clarified in the methods section (Line 281-282) and in the timeline of the Figure 1A

5. Typos in the text and the graphs (EV2 and : closAed arms)

Typos have been corrected

Prof. Nicolas Toni
Lausanne University/Lausanne University Hospital
Center for Psychiatric Neurosciences
Prilly 1008
Switzerland

Dear Nicolas, and happy new year !

I am very pleased to accept your manuscript for publication in the next available issue of EMBO reports. Thank you for your contribution to our journal.

Best wishes,
Esther
